# scMRDR: A Scalable and Flexible Framework for Unpaired Single-Cell Multi-Omics Data Integration

**Jianle Sun**[1,2], **Chaoqi Liang**[1], **Ran Wei**[1], **Peng Zheng**[1],
**Lei Bai**[1], **Wanli Ouyang**[1], **Hongliang Yan**[4], **Peng Ye**[1,3†]

[1] Shanghai Artificial Intelligence Laboratory, [2] Carnegie Mellon University,
[3] The Chinese University of Hong Kong, [4] Guangzhou Laboratory

## Abstract

Advances in single-cell sequencing have enabled high-resolution profiling of diverse molecular modalities, while integrating unpaired multi-omics single-cell data remains challenging. Existing approaches either rely on pair information or prior correspondences, or require computing a global pairwise coupling matrix, limiting their scalability and flexibility. In this paper, we introduce a scalable and flexible generative framework called single-cell Multi-omics Regularized Disentangled Representations (scMRDR) for unpaired multi-omics integration. Specifically, we disentangle each cell's latent representations into modality-shared and modality-specific components using a well-designed $\beta$-VAE architecture, which are augmented with isometric regularization to preserve intra-omics biological heterogeneity, adversarial objective to encourage cross-modal alignment, and masked reconstruction loss strategy to address the issue of missing features across modalities. Our method achieves excellent performance on benchmark datasets in terms of batch correction, modality alignment, and biological signal preservation. Crucially, it scales effectively to large-scale datasets and supports integration of more than two omics, offering a powerful and flexible solution for large-scale multi-omics data integration and downstream biological discovery.

## 1 Introduction

Recent advances in single-cell sequencing technologies have enabled the measurement of diverse molecular modalities at single-cell resolution, such as gene expression (scRNA), chromatin accessibility (scATAC), and protein abundance (scProtein). These complementary data sources offer a comprehensive view of cellular states and dynamics. Although a few protocols allow limited joint profiling using marker-based techniques, they still suffer from low feature coverage and reduced flexibility due to the destructive nature of single-cell assays, making it remain technically challenging to jointly measure multiple modalities within the same cell. Consequently, large-scale single-cell datasets are typically unpaired across different modalities [33]. This unpaired nature, coupled with significant technical noise such as batch effects, dropouts, and sequencing depth variation, makes data integration in a shared biologically meaningful latent space a highly nontrivial task [27, 3].

The goal of multi-omics data integration is to map single-cell data in different omics into a shared latent space, where representations across modalities are distributionally aligned while preserving biological differences between cell types and correcting for technical variations due to experimental batches (Fig.1a). Existing approaches explored joint dimension reduction (Fig.1b), like factor

---

†Corresponding author.
Source codes are available at `https://github.com/sjl-sjtu/scMRDR`.
Email Address: jianles@andrew.cmu.edu. Work was done during an internship at Shanghai AI Laboratory.

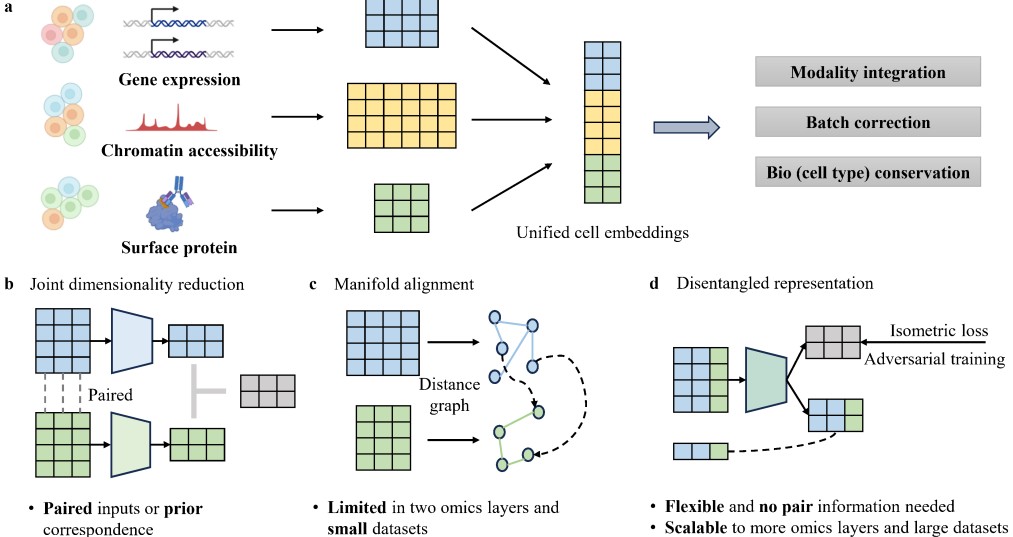

Figure 1: Method overview. (a) Multi-omics data integration. The goal is to integrate single-cell data in different modalities into an aligned latent space while preserving biological information and correcting technical noise. (b) Integration via joint dimension reduction (e.g., joint autoencoders). It typically works with paired data (measurements on different omics within the same cell). (c) Integration via manifold alignment between the geometric structures (e.g., KNN distance graphs) of different omics. It does not require paired data, but is typically limited to small-scale datasets involving only two omics modalities. (d) Our framework, based on disentangled representations, is flexible to completely unpaired data and scalable to large datasets with more than two omics.

analysis [1], or deep generative models [24]. However, they typically rely on paired or partially paired data [2, 6] to guide the integration, or require external prior knowledge [8] or pre-learned coupling matrix [11] to bridge modalities, limiting their **flexibility**. On the other hand, some approaches employ manifold alignment (Fig.1c), including optimal transport [14, 7] or unsupervised manifold transformation [5, 39]. However, these methods, relying on computing a global pairwise coupling matrix, typically restrict to integrating two modalities, and encounter serious computational issues in large datasets due to the complexity of inter-modal alignment, struggling in **scalability**.

To address the challenges, we proposed a scalable and flexible generative model named single-cell Multi-omics Regularized Disentangled Representations (scMRDR) to integrate unpaired multi-omics single-cell data into a unified latent space (Fig.1d). Unlike existing methods, scMRDR requires neither paired samples and prior information nor establishing global correspondences across different modalities. Instead, we achieve the integration based on the disentanglement of each sample's latent code into *shared* and *modality-specific* components via a well-designed $\beta$-VAE architecture, incorporating *isometric regularization* to ensure the conservation of biological information, and *adversarial training* to encourage the fusion of different modalities, with *masked loss function* to address the feature missing issue in different modalities (Fig.2).

Due to the *single unified encoder-decoder architecture*, scMRDR is flexible to completely unpaired data and able to scale to large datasets with more than two modalities naturally. Applied to real-world unpaired single-cell data, scMRDR demonstrates excellent performance in modality integration, batch correction as well as bio-conservation, surpassing a broad range of existing methods. Moreover, scMRDR scales robustly to large datasets, and readily accommodates additional omics layers. These results collectively underscore scMRDR as a flexible and scalable framework for large-scale unpaired single-cell data integration and the discovery of complex biological mechanisms.

We summarize the contributions as follows: (1) Through in-depth analysis of existing works in the field of multi-omics integration, we identify their limitations in flexibility and scalability, and propose a generative framework with a unified single encoder-decoder $\beta$-VAE to disentangle latent representations; (2) We propose a joint optimization goal, incorporating isometric regularization, adversarial

training, and masked loss to facilitate modality fusion while preserving biological signals; (3) We validate scMRDR through extensive experiments on multiple real-world datasets, demonstrating its strong flexibility and scalability on large-scale datasets and more complex multi-omics integration tasks, as well as its practical significance in downstream biological analyses (such as spatial position imputation).

## 2 Related work

Integrating multi-omics data has been extensively studied, with methods typically falling into two broad categories. Joint dimension reduction methods, including statistical models such as factor analysis based method like MOFA [1], canonical correlation analysis (CCA)-based methods like Seurat v3 [30], as well as deep generative models such as scVI-based [24] adaptations, assume access to matched (paired) measurements across omics layers, enabling supervised or semi-supervised learning of joint representations. For example, MultiVI [2] integrates paired multi-omic data by directly averaging latent embeddings inferred by encoders of respective modalities. However, the need for explicit pairing limits their applicability in cases where cross-modality correspondences are incomplete or noisy. JAMIE [11] incorporates the manifold alignment approach into the VAE framework, and UniPort [6] takes advantage of partially paired features and coupled VAE, but they still confront computational intensity in dealing with large-scale clinical or experimental datasets.

Another prominent line of work aligns modalities through unsupervised manifold matching, optimizing for geometric consistency between latent spaces. UnionCom [5] aligns modalities by constructing a kNN graph and applying unsupervised linear manifold alignment, while CMOT [39] adopts non-linear manifold alignment with partial supervision. SCOT [14] leverages Gromov-Wasserstein optimal transport (GWOT) on similarity matrices and uses barycentric projection for integration, with following revised versions such as Pamona [7] and SCOTv2 [13] by imposing regularizations on GWOT, while SCOOTR [15] aligns both samples and features using co-optimal transport (COOT). These methods typically require constructing pairwise similarities and a global coupling matrix, restricting scalability to large datasets due to computational bottlenecks, and in practice, they are often validated only on limited sample sizes or toy examples. Moreover, they often focus on the integration of two modalities, leaving the integration of more omics layers an underexplored challenge.

Recent works, such as scTFBridge [34], scMaui [20], and InClust+ [35], have discussed integration based on latent decomposition. However, they still rely on designs like partial pairing supervision, stacked encoders, and cross-modal contrastive learning, limiting their scalability in completely unpaired and multi-omics contexts. In contrast, we aim to achieve the decoupling via a unified $\beta$-VAE composed of a single encoder-decoder, treating observations in different omics equally as a single sample, thereby ensuring flexibility and scalability in completely unpaired data across multiple omics. Theoratically, such disentangled subspaces are unidentifiable (i.e., not unique) without additional constraints [21]. We leverage this unidentifiability and, by imposing isometric and adversarial regularization, constrain the modality-shared subspace to be the one that preserves the maximum sample structure information from the entire space while aligning different modalities.

## 3 Methods

### 3.1 Preliminary: Disentangled VAE

Variational Autoencoders (VAEs) [22] are a class of generative models that learn a probabilistic mapping between observed data $x$ and latent variable space $z$ via variational inference by introducing an encoder network to parametrize the variational posterior $q_\phi(z \mid x)$ and a decoder network to reconstruct the generative process $p_\theta(x \mid z)$. The model is trained by maximizing the evidence lower bound (ELBO) on the marginal log-likelihood:

$$\text{ELBO}_{\text{VAE}}(x) = \mathbb{E}_{q_\phi(z|x)}[\log p_\theta(x \mid z)] - \text{KL}(q_\phi(z \mid x) \parallel p(z)), \tag{1}$$

where $\text{KL}(q \parallel p)$ denotes the Kullback–Leibler (KL) divergence between distributions $q$ and $p$. The first term encourages faithful data reconstruction, while the second regularizes the latent space to align with the prior.

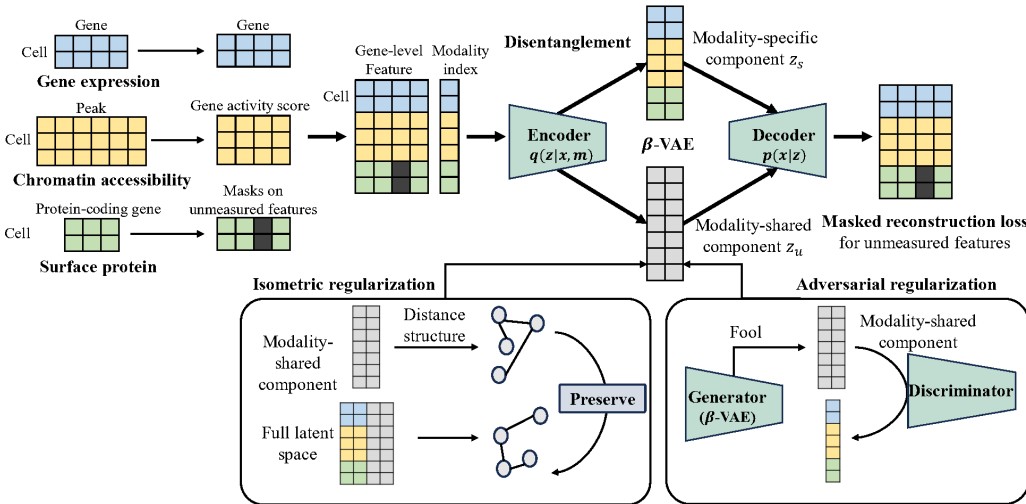

Figure 2: Overview of the proposed scMRDR. We employ $\beta$-VAE to disentangle omics-specific and omics-shared latent representations, and impose isometric loss and adversarial training as regularization to encourage modality integration and bio-conservation.

Classical VAE often conflates modality-shared and modality-specific signals in the latent space, impeding interpretability and downstream analyses. To improve disentanglement in the learned latent representations, $\beta$-VAE [19] introduces a hyperparameter $\beta > 1$ to upweight the KL divergence term in the VAE objective

$$\text{ELBO}_{\beta\text{-VAE}}(x) = \mathbb{E}_{q_\phi(z|x)}[\log p_\theta(x \mid z)] - \beta \cdot \text{KL}(q_\phi(z \mid x) \parallel p(z)). \tag{2}$$

A higher value of $\beta$ enforces a stronger constraint on the latent space, promoting disentangled and interpretable representations at the cost of reduced reconstruction accuracy.

Simply disentangling the latent space does not guarantee that the shared latent components from different omics data are fully aligned in distribution, nor does it ensure that the shared latent components preserve the structural information present in the original data. Such structural information is captured by the VAE in the entire latent space and may not necessarily be retained within the shared subspace. We will impose additional regularization on the disentangled latent space in a generative model designed for single-cell multi-omics to address the issue.

### 3.2 Disentangled generative model for multi-omics data

To achieve flexible and scalable integration, we propose a generative model tailored for single-cell multi-omics data (Fig.3). We assume that observations $x^{(m)}$ in the omics $m$ are generated from latent embeddings lying in two independent subspaces, i.e., common latent variables $z_u$ shared across modalities and modality-specific latent variables $z_s^{(m)}$, and we have

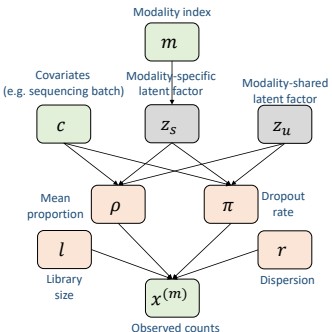

Figure 3: Graphical illumination of the single-cell multi-omics data generative model.

$$p(z|m) = p(z_u)p(z_s|m) \tag{3}$$

and

$$p(x) = p(x|z,c)p(z|m)p(m)p(c) = p(x|z,c)p(z_u)p(z_s|m)p(m)p(c) \tag{4}$$

where $c$ represents other covariates, such as the experimental batch during sequencing. Batch effects are systematic variations introduced by non-biological factors, including differences in experimental runs, reagent lots, or operators. These effects can obscure true biological signals or introduce spurious

patterns in the data [32]. To mitigate such confounding influences, batch information is commonly included as a covariate in the modeling process.

Sequencing reads of scRNA, protein, and other omics data can all be mapped to the gene level (e.g., gene activity scores derived from peak aggregation in single-cell ATAC-seq). Since the read matrix are typically sparse (with both biological and technical dropouts) count data with over-dispersion (i.e., the variance exceeds the mean), we parametrize the generative process $p(x|z, c)$ using a zero-inflated negative binomial (ZINB) distribution [24, 29] (which can be changed to ordinary Gaussian model if normalized scores or more non-count data included), i.e.,

$$x_{ng} \sim \pi_g \delta_0 + (1 - \pi_{ng})\text{NB}(l_n \rho_{ng}, r_g) \overset{d}{=} \text{ZINB}(l_n \rho_{ng}, r_g, \pi_g) \tag{5}$$

where $\delta_0$ is point mass at zero, $l_n$ represents the library size of cell $n$, $\rho_{ng}$ is the mean proportion of the corresponding measurement (RNA expressions, activity score, protein level, etc.) of gene $g$ in cell $n$, $r_g$ is the dispersion factor of gene $g$, $\pi_{ng}$ is the dropout rate of gene $g$ in cell $n$. We parametrize the parameters by some non-linear neural networks as follows $h = f_h(z, c)$, $\rho_{ng} = f_\rho(h)$, $\pi_{ng} = f_\pi(h)$.

Prior distributions of latent factors $z$ are assumed as isotropic multivariate Gaussian distributions, i.e., $p(z_s|m) \overset{d}{=} \mathcal{N}(\mu_m, \sigma_m^2 I)$ and $p(z_u|t) \overset{d}{=} \mathcal{N}(0, I)$. We employ variational posteriors $q(z_u|x) = \mu_u(x) + \sigma_u(x) \odot \mathcal{N}(0, I)$ and $q(z_s|x, m) = \mu_s(x, m) + \sigma_s(x, m) \odot \mathcal{N}(0, I)$ to approximate the prior, and the loss function (negative ELBO) of $\beta$-VAE is

$$\begin{aligned}
\mathcal{L}_{\beta\text{-VAE}} &= -\text{ELBO} = \mathcal{L}_{\text{recon}} + \beta \mathcal{L}_{\text{KL}} \\
&= -\mathbb{E}_{z \sim q(z|x,m)} \log p(x|z, c) + \beta \left[ \text{KL}(q(z_u|x)\|p(z_u)) + \text{KL}(q(z_s|x, m)\|p(z_s|m)) \right]
\end{aligned} \tag{6}$$

where $\beta > 1$ to encourage the disentanglement of $z_u$ and $z_s$. For the ZINB model, the reconstruction loss, i.e., the expected log likelihood under the variational posterior, is

$$\mathbb{E}_{z \sim q(z|x,m)} \log p(x|z, c) = \frac{1}{N} \sum_{n=1}^{N} \sum_{g=1}^{G} \log \left[ P_{\text{ZINB}}(x_n; l_n \rho_{ng}, r_g, \pi_{ng}) \right] \tag{7}$$

where $P_{\text{ZINB}}(X = x; \mu, r, \pi) = \pi \mathbb{I}_{x=0} + (1 - \pi) P_{\text{NB}}(x; \mu, r)$, and $P_{\text{NB}}(x; \mu, r)$ stands for the probability mass of negative binomial distribution $NB(\mu, r)$ at $x$, i.e., $P_{\text{NB}}(x; \mu, r) = \frac{\Gamma(x+r)}{x! \, \Gamma(r)} \left( \frac{r}{r+\mu} \right)^r \left( \frac{\mu}{r+\mu} \right)^x$. In particular, the probability mass at zero $P_{\text{NB}}(0; \mu, r) = (\frac{r}{r + \mu})^r$.

### 3.3 Adversarial regularization for omics integration

By disentangling the latent space, we obtain, in general, modality-invariant latent variable $z_u$ lying in a shared subspace. To further encourage the alignment of distributions $z_u^{(m)}$ from different omics, we impose an additional adversarial regularization [17, 16] by introducing a $m$-class discriminator $D(z_u) : \mathbb{R}^{d_u} \to \{0, 1, \dots, m\}$ to distinguish $z_u$ of samples from different omics and try to optimize its capacity, i.e.,

$$\min_D \mathcal{L}_{\text{discriminator}} = \min_D \left[ -\sum_m m \log(D(z_u^{(m)})) \right], \quad z_u^{(m)} \sim q(z_u|x) \tag{8}$$

while training the VAE encoders $q(z_u|x)$ to fool the discriminator as much as possible by optimizing in the opposite direction

$$\max_{q(z_u|x)} \inf_D \mathcal{L}_{\text{discriminator}} \tag{9}$$

which is equivalent to

$$\min_{q(z_u|x)} \mathcal{L}_{\text{alignment}} = \min_{q(z_u|x)} \sup_D \left[ \sum_m m \log \left( D(z_u^{(m)}) \right) \right] \tag{10}$$

achieving a proper alignment of embeddings from different omics ultimately.

### 3.4 Isometric loss for structure preservation

To ensure that $z_u$ captures the biological differences between samples (e.g., cell types), we introduce an additional unsupervised structure-preserving regularization, since cell type labels are typically

unavailable. Though the original feature matrices are high-dimensional, the full latent representation $z = (z_u, z_s)$ learned by the generative model effectively preserves intra-modality structure. Hence, we reformulate the problem as encouraging $z_u$ to retain the structural information of the full latent space. Specifically, since the latent embeddings already reside in a low-dimensional space, we apply an isometric loss [31] that minimizes the discrepancy between the pairwise Euclidean distance matrices computed from $z$ and from $z_u$, for each modality, i.e.,

$$\mathcal{L}_{\text{preserve}} = \sum_m \sum_{i,j \in X^{(m)}} \left[ ||\mu_{z_u}(x_i) - \mu_{z_u}(x_j)||_2 - ||\mu_z(x_i) - \mu_z(x_j)||_2 \right]^2 \tag{11}$$

where $\mu_{z_u}(x)$ is the posterior mean of variational approximation $q(z_u|x)$ and $\mu_z(x)$ is the posterior mean of total latent embeddings $(q(z_u|x), q(z_s|x, m))$.

And the total optimization goal for VAE becomes

$$\mathcal{L}_{\text{total}} = \mathcal{L}_{\text{recon}} + \beta \mathcal{L}_{\text{KL}} + \lambda \mathcal{L}_{\text{alignment}} + \gamma \mathcal{L}_{\text{preserve}} \tag{12}$$

In the training process, we first update the discriminator by optimizing $\mathcal{L}_{\text{discriminator}}$, then update VAE with respect to the total loss $\mathcal{L}_{\text{total}}$ in turn in each mini-batch.

### 3.5 Masked reconstruction loss for missing features

In unpaired multi-omics datasets, different modalities are measured separately and often originate from distinct sources. Although it is possible to align features across modalities at the gene level, severe missing features still exist due to different sequencing coverages, especially for antibody-based protein profiling techniques such as CITE-seq, which typically covers only a few hundred proteins due to the limited availability of antibody markers [4], while tens of thousands of genes can be measured in other omics. Restricting the analysis to the overlapping features across all modalities would lead to substantial information loss, whereas naively imputing unmeasured features with zeros would severely distort the data distribution. To address this, we introduce a binary mask $\mathbf{b} \in \{0, 1\}^G$ indicating feature availability that prevents gradients from back-propagating through unmeasured features in the reconstruction loss for each modality, and then scale by the proportion of available features to ensure that the reconstruction loss for each sample is on a comparable scale, i.e.,

$$\mathcal{L}_{\text{recon}} = -\frac{1}{N} \sum_{n=1}^{N} \left\{ \frac{G}{\sum_{g=1}^{G} \mathbf{b}_{ng}} \sum_{g=1}^{G} \mathbf{b}_{ng} \log \left[ P_{\text{ZINB}}(x_n; l_n \rho_{ng}, r_g, \pi_{ng}) \right] \right\} \tag{13}$$

The masked loss strategy ensures that the model can fully utilize the available information while preserving the integrity of the original data distribution.

## 4 Results

### 4.1 Setup and evaluation metrics

To verify the effectiveness of our proposed method, we comprehensively evaluate scMRDR through a series of experiments, beginning with standard benchmarks and then scaling up to more complex single-cell and multi-omics scenarios. We employ publicly available datasets from previous researches with curated cell-type annotations [28, 37, 25]. Detailed setups are shown in Appendix A.1 and Table 2. We compare to state-of-the-art baselines, including GLUE, scVI, Seurat v5, Harmony, JAMIE, and so on, and evaluate in terms of cell-type clustering, modality integration, and batch removal. Cell-type labels in different omics has been aligned in evaluation. It should be emphasized that we did not use cell-type labels during training. UMAP visualizations are presented for qualitative comparison. For quantitative evaluation, the following commonly used metrics (Appendix A.2) are included: F1 isolated label scores, k-means NMI, k-means ARI, cell-type Silhouette, and cell-type separation LISI (cLISI) to evaluate the performance in cell type conservation, modality Silhouette, modality integration LISI (iLISI), kBET, Principal Component Regression (PCR) $R^2$, and graph connectivity to evaluate the performance in modality integration, as well as batch Silhouette, batch integration LISI (iLISI), kBET, and PCR $R^2$ to evaluate the performance in batch effect correction [26, 36].

We evaluate and visualize all the above metrics based on the package scib-metrics [26]. The overall score is calculated as a weighted average of all metrics on bio-conservation (40% weights), modality integration (30% weights), and batch correction (30% weights).

Figure 4: Performance comparisons on two-omics integration, where unscaled metrics calculated via scIB are reported.

| Method | Bio conservation | | | | | Batch correction | | | | Modality integration | | | | | Aggregate score | | | |
|---|---|---|---|---|---|---|---|---|---|---|---|---|---|---|---|---|---|---|
| | Isolated labels | KMeans NMI | KMeans ARI | Silhouette label | cLISI | Silhouette batch | iLISI | KBET | PCR comparison | Silhouette modality | iLISI | KBET | Graph connectivity | PCR comparison | Batch correction | Bio conservation | Modality integration | Total |
| Ours | 0.69 | 0.76 | 0.58 | 0.66 | 1.00 | 0.90 | 0.52 | 0.38 | 0.26 | 0.86 | 0.37 | 0.32 | 0.96 | 0.99 | 0.52 | 0.78 | 0.70 | 0.66 |
| GLUE | 0.65 | 0.77 | 0.57 | 0.67 | 1.00 | 0.90 | 0.42 | 0.28 | 0.09 | 0.85 | 0.60 | 0.34 | 0.94 | 0.99 | 0.42 | 0.73 | 0.74 | 0.64 |
| scVI | 0.59 | 0.68 | 0.43 | 0.56 | 1.00 | 0.95 | 0.47 | 0.29 | 0.37 | 0.81 | 0.00 | 0.00 | 0.71 | 0.85 | 0.52 | 0.65 | 0.47 | 0.56 |
| MaxFuse | 0.65 | 0.73 | 0.49 | 0.59 | 1.00 | 0.89 | 0.16 | 0.19 | 0.00 | 0.87 | 0.15 | 0.19 | 0.91 | 1.00 | 0.31 | 0.69 | 0.62 | 0.56 |
| Seurat | 0.68 | 0.78 | 0.61 | 0.68 | 1.00 | 0.90 | 0.06 | 0.19 | 0.00 | 0.70 | 0.00 | 0.34 | 0.46 | 0.99 | 0.29 | 0.75 | 0.50 | 0.54 |
| Pamona | 0.50 | 0.22 | 0.18 | 0.50 | 0.96 | 0.93 | 0.51 | 0.26 | 0.61 | 0.74 | 0.00 | 0.09 | 0.43 | 1.00 | 0.58 | 0.47 | 0.45 | 0.50 |
| JAMIE | 0.43 | 0.30 | 0.17 | 0.43 | 1.00 | 0.86 | 0.26 | 0.27 | 0.81 | 0.56 | 0.00 | 0.08 | 0.45 | 0.99 | 0.55 | 0.47 | 0.42 | 0.48 |
| SIMBA | 0.49 | 0.01 | 0.01 | 0.49 | 0.76 | 0.98 | 0.70 | 0.27 | 0.95 | 0.90 | 0.00 | 0.00 | 0.03 | 0.83 | 0.73 | 0.35 | 0.35 | 0.46 |
| Harmony | 0.54 | 0.60 | 0.41 | 0.54 | 1.00 | 0.89 | 0.19 | 0.27 | 0.00 | 0.56 | 0.00 | 0.06 | 0.59 | 0.64 | 0.34 | 0.62 | 0.37 | 0.46 |
| UnionCom | 0.42 | 0.43 | 0.08 | 0.44 | 0.98 | 0.81 | 0.43 | 0.39 | 0.00 | 0.38 | 0.00 | 0.01 | 0.41 | 1.00 | 0.41 | 0.47 | 0.36 | 0.42 |

| Method | Bio conservation | | | | | Batch correction | | | | Modality integration | | | | | Aggregate score | | | |
|---|---|---|---|---|---|---|---|---|---|---|---|---|---|---|---|---|---|---|
| | Isolated labels | KMeans NMI | KMeans ARI | Silhouette label | cLISI | Silhouette batch | iLISI | KBET | PCR comparison | Silhouette modality | iLISI | KBET | Graph connectivity | PCR comparison | Batch correction | Bio conservation | Modality integration | Total |
| Ours | 0.58 | 0.58 | 0.37 | 0.56 | 1.00 | 0.90 | 0.24 | 0.14 | 0.97 | 0.82 | 0.26 | 0.10 | 0.95 | 1.00 | 0.56 | 0.62 | 0.62 | 0.60 |
| Seurat | 0.61 | 0.69 | 0.51 | 0.61 | 1.00 | 0.86 | 0.18 | 0.09 | 0.98 | 0.77 | 0.01 | 0.07 | 0.80 | 1.00 | 0.53 | 0.68 | 0.53 | 0.59 |
| GLUE_lsi | 0.53 | 0.55 | 0.33 | 0.53 | 0.99 | 0.90 | 0.23 | 0.09 | 0.98 | 0.52 | 0.10 | 0.11 | 0.75 | 1.00 | 0.55 | 0.59 | 0.49 | 0.55 |
| scVI | 0.53 | 0.57 | 0.26 | 0.52 | 1.00 | 0.94 | 0.12 | 0.10 | 0.93 | 0.72 | 0.00 | 0.00 | 0.82 | 0.95 | 0.52 | 0.57 | 0.50 | 0.54 |
| Harmony | 0.52 | 0.47 | 0.21 | 0.51 | 1.00 | 0.93 | 0.10 | 0.10 | 0.67 | 0.57 | 0.00 | 0.00 | 0.58 | 0.68 | 0.45 | 0.54 | 0.37 | 0.46 |
| GLUE_pca | 0.44 | 0.38 | 0.21 | 0.42 | 0.99 | 0.70 | 0.14 | 0.09 | 0.95 | | 0.00 | 0.00 | 0.48 | 0.96 | 0.47 | 0.49 | 0.37 | 0.45 |
| MaxFuse | 0.42 | 0.26 | 0.01 | 0.40 | 0.97 | 0.66 | 0.13 | 0.05 | 0.99 | 0.66 | 0.00 | 0.01 | 0.45 | 1.00 | 0.46 | 0.41 | 0.42 | 0.43 |
| SIMBA | 0.49 | 0.01 | 0.00 | 0.49 | 0.73 | 0.98 | 0.12 | 0.10 | 0.85 | 0.89 | 0.00 | 0.09 | 0.07 | 0.85 | 0.51 | 0.35 | 0.38 | 0.41 |

Figure 5: Performance comparisons on two-omics integration with large-scale dataset, where unscaled metrics calculated via scIB are reported. The default preprocessing method 'scglue.data.lsi' for GLUE fails to handle the large-scale data, and substituting it with PCA leads to severe performance degradation, although using 'TruncatedSVD' as an approximation of LSI can alleviate this issue.

## 4.2 Benchmarking performance on two-omics integration

We first compare scMRDR with 9 existing methods, including Seurat v5 [18], Harmony [23], scVI [24], scGLUE [8], JAMIE [11], UnionCom [5], Pamona [7], MaxFuse [10] and SIMBA [9] on a unpaired scRNA and scATAC dataset of human kidney tissue [28]. Among these, scVI can be regarded as a baseline counterpart of our model, with no disentanglement or regularization applied to the latent space. The dataset contains scRNA-seq with 27,146 genes on 19,985 cells and scATAC-seq with 99,019 peaks on 24,205 cells. Peak signals in scATAC are aggregated to gene-level activity score in scMRDR by the package episcanpy [12]. Gene activity scores are also used in integration by Seurat, Harmony, and scVI, while others use raw peak signals directly. We choose the highly variable genes for each omics and take the union as input. Shown in Fig.4, scMRDR outperforms the existing methods, exhibiting an excellent performance in modality integration, bio-conservation, and batch correction. As shown in Fig.6a, without the incorporation of explicit cell type annotations in training, our method yields well-separated embedding clusters corresponding to distinct cell types in an unsupervised way, thereby preserving the underlying biological heterogeneity. Meanwhile, samples from different omics and batches fuse and align well in the latent space, demonstrating successful correction of modality-specific variations and technical noises like batch effects. In contrast, some other methods such as Harmony, scVI and JAMIE (Fig.9 in Appendix A.5) can preserve biological differences between distinct cell types, but fail to integrate the distributions of the two different omics modalities.

## 4.3 Scalability on integration with large-scale dataset

To validate the scalability of scMRDR on larger datasets, we evaluate the performance on an large-level dataset on mouse primary motor cortex with more cells available [37]. This large-scale data includes 69,727 cells with measurements on 27,123 genes by scRNA-seq and 54,844 cells with measurements on 148,814 chromatin peaks by scATAC-seq. Methods based on optimal transport or other unsupervised manifold alignment, such as JAMIE, UnionCom, and Pamona, fail to run on datasets with such scale due to errors in memory or optimization. We compare the performance of the rest methods. Shown in Fig.5, some methods that perform well on small-scale datasets suffer significant performance drops on larger ones (Fig.10 in Appendix A.5). For example, GLUE exhibits

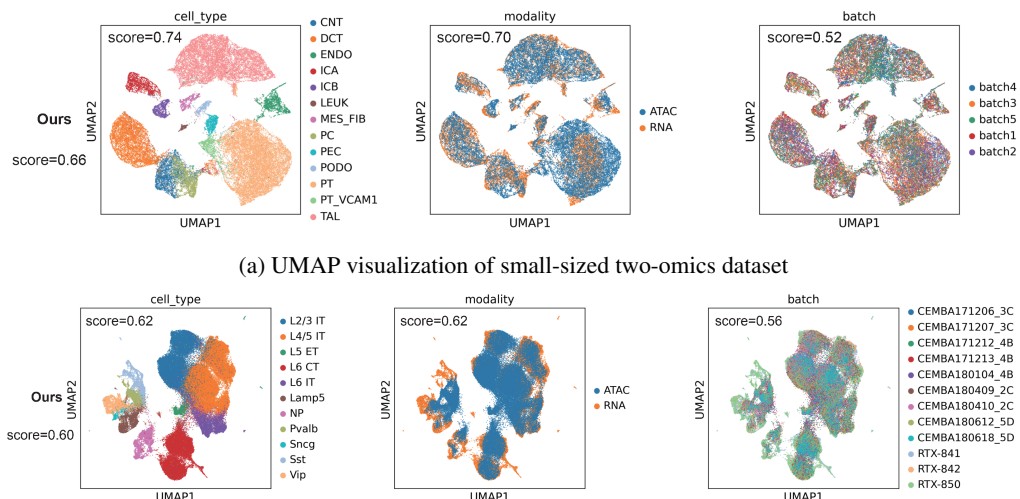

(a) UMAP visualization of small-sized two-omics dataset

(b) UMAP visualization of large-scale two-omics dataset

Figure 6: UMAP visualization of the latent representations obtained by scMRDR in the two-omics integration task. Latent embeddings of other methods are shown in Appendix A.5. An effective method should yield well-separated clusters corresponding to distinct cell types, and fuse samples from different modalities and experimental batches in sequencing. Noted that the annotated cell type labels are not incorporated in the unsupervised learning but only used in evaluation.

| Method | Bio conservation | | | | | Batch correction | | | | Modality integration | | | | | Aggregate score | | | |
|---|---|---|---|---|---|---|---|---|---|---|---|---|---|---|---|---|---|---|
| | Isolated labels | KMeans NMI | KMeans ARI | Silhouette label | cLISI | Silhouette batch | iLISI | KBET | PCR comparison | Silhouette modality | iLISI | KBET | Graph connectivity | PCR comparison | Batch correction | Bio conservation | Modality integration | Total |
| **Ours** | 0.57 | 0.68 | 0.58 | 0.61 | 1.00 | 0.83 | 0.27 | 0.16 | 0.31 | 0.83 | 0.28 | 0.06 | 0.87 | 1.00 | 0.39 | 0.69 | 0.61 | 0.58 |
| **GLUE** | 0.55 | 0.67 | 0.55 | 0.59 | 1.00 | 0.86 | 0.20 | 0.13 | 0.25 | 0.85 | 0.15 | 0.09 | 0.82 | 0.99 | 0.36 | 0.67 | 0.58 | 0.55 |
| **scVI** | 0.53 | 0.50 | 0.31 | 0.54 | 1.00 | 0.90 | 0.20 | 0.32 | 0.53 | 0.76 | 0.00 | 0.00 | 0.46 | 0.70 | 0.49 | 0.58 | 0.38 | 0.49 |
| **Harmony** | 0.45 | 0.35 | 0.16 | 0.44 | 1.00 | 0.76 | 0.17 | 0.11 | 0.12 | 0.51 | 0.00 | 0.00 | 0.40 | 0.00 | 0.29 | 0.48 | 0.18 | 0.33 |

Figure 7: Performance comparisons on triple-omics integration, where unscaled metrics calculated via scIB are reported.

strong susceptibility to the choice of preprocessing strategy which itself markedly dependent on data scale, whereas our method maintains stable performance on large-scale datasets (Fig.6b). This makes it highly scalable for large-level single-cell data integration, enabling the informative linking among different omics and providing more valuable biological insights.

## 4.4 Scalability on triple-omics integration

Most existing methods do not support integrating datasets with more than two omics, while scMRDR can naturally extend to the integration of triple-omics or even more. To illuminate it, we conduct a case study on integrating scRNA, scATAC, and CITE-seq measured sc-protein levels, using a dataset on human bone marrow [25]. We conducted integration on 30,486 cells with scRNA-seq on 13,431 genes, 10,330 cells with scATAC-seq on 116,490 peaks, and 18,052 cells with 134 surface proteins measured by CITE-seq. Some methods that perform relatively well on two-omics datasets, such as Seurat v5, are not suitable for integrating more modalities, as they require dictionary learning and bridge integration between two omics. As a result, we compare the performance of the existing methods that support triple-omics integration, including GLUE, scVI, and Harmony. Shown in Fig.7, scMRDR shows a consistently excellent performance in the integration task with more modalities On the contrary, methods like GLUE fail to align latent distributions over three omics, especially when one of the omics modalities (proteomics here) has significantly fewer measured features than the others (Fig.8).

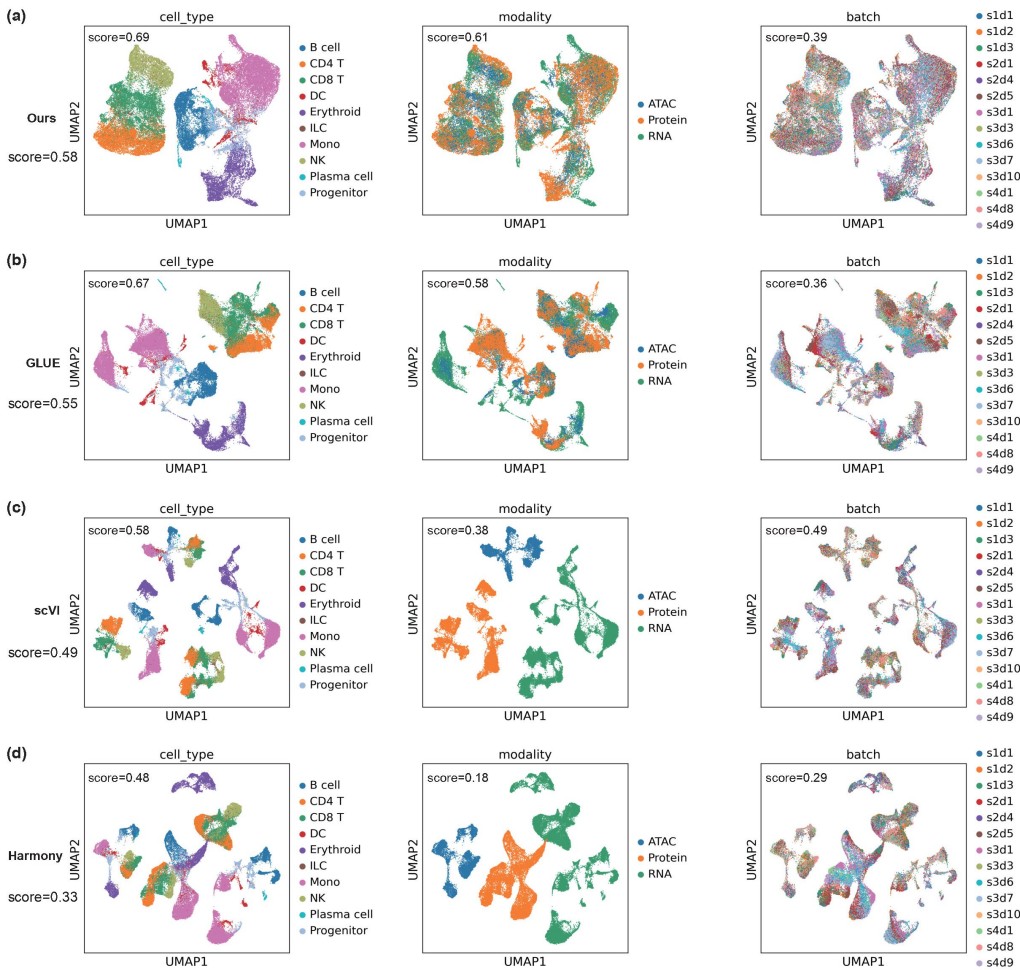

Figure 8: UMAP visualization of the unified embeddings in triple-omics data.

## 4.5 Ablation study and sensitivity analysis on the regularized beta-VAE

To demonstrate the effects of isometric and adversarial regularization within the $\beta$-VAE framework, we conducted ablation experiments and sensitivity analysis with a range of different hyperparameter combinations on the two-omics human kidney datasets. Shown in Table 1, with different hyperparameters, scMRDR performs consistently better than the ablation results when removing any individual component, and the worst model is the baseline without any regularization. The absence of isometric constraints or modality-adversarial regularization leads to a substantial drop in performance, where isometry is more essential to bio-conservation while adversary more vital to modality alignment. The influence of $\beta$ is relatively minor as the structured prior imposed on $z$ inherently promotes disentanglement. Nevertheless, employing a moderately large $\beta$ can increase the conditional independence between $z_u$ and $z_s$, thereby enhancing the effectiveness of latent disentanglement. While different hyperparameter combinations have a certain impact due to trade-offs among various losses and the influence of randomness, the performance are generally robust to the choices.

We also conduct ablation studies on a triple-omics dataset to evaluate the effectiveness of the masked loss. The results (Table 4 in Appendix A.4) show that, keeping all other hyperparameters unchanged, simply setting the missing features in the proteomics to zero without applying a loss mask leads to a significant performance drop. This highlights the importance of our masked loss strategy in effectively handling integration tasks where some omics layers (e.g., scProtein) contain substantially fewer features compared to other modalities.

Table 1: Performance in ablation studies and sensitivity analysis, where unscaled scIB aggregate scores are reported

| $\beta$ | $\gamma$ | $\lambda$ | Total Score | Batch-correct | Bio-conserve | Modal-integrate | Notes |
|---|---|---|---|---|---|---|---|
| 2 | 5 | 5 | 0.66 | 0.52 | 0.74 | 0.70 | |
| 2 | 7 | 5 | 0.65 | 0.54 | 0.69 | 0.72 | |
| 2 | 5 | 3 | 0.65 | 0.55 | 0.69 | 0.71 | |
| 2 | 2 | 2 | 0.65 | 0.50 | 0.72 | 0.70 | |
| 2 | 5 | 7 | 0.65 | 0.49 | 0.70 | 0.73 | |
| 2 | 3 | 5 | 0.65 | 0.47 | 0.71 | 0.74 | |
| 2 | 1 | 5 | 0.64 | 0.49 | 0.68 | 0.75 | |
| 3 | 5 | 5 | 0.64 | 0.48 | 0.71 | 0.71 | |
| 4 | 5 | 5 | 0.64 | 0.47 | 0.71 | 0.72 | |
| 2 | 10 | 5 | 0.64 | 0.52 | 0.68 | 0.71 | |
| 1 | 5 | 5 | 0.62 | 0.50 | 0.67 | 0.68 | $\beta = 1$ |
| 2 | 5 | 0 | 0.61 | 0.53 | 0.69 | 0.60 | $\lambda = 0$ |
| 2 | 0 | 5 | 0.61 | 0.51 | 0.60 | 0.72 | $\gamma = 0$ |
| 1 | 0 | 0 | 0.59 | 0.48 | 0.63 | 0.66 | Baseline |

### 4.6 Biological significance of integrating single-cell and spatial omics

To better evaluate the biological significance of scMRDR, we integrated scRNA [37], scATAC [37], and spatial transcriptomics (merFISH) [38] of mouse primary motor cortex using our method, and then used the aligned latent representation to interpolate the missing spatial locations in single-cell data by optimal transport. Visualization shows that this interpolation performs well, where inferred locations of cells align well with the provided cortex layers annotations (Fig. 12 in Appendix).

Due to the low coverage of merFISH (only 254 genes measured), only 103 genes are detected as spatial variable genes (SVGs) by SPARKX ($P_{\text{adj}} < 10^{-20}$). We leveraged the spatially interpolated scRNA data (26069 genes) and 4095 SVGs ($P_{\text{adj}} < 10^{-20}$) are detected. We replicated 83 out of 101 SVGs detectable by merFISH (like *Lamb5*, *Calb1*, *Cux2*), and also revealed new SVGs (like *Hs3st4*, *Cpa6*, *Zfhx4*). Similarly, using scATAC with imputed spatial locations, we identified 142 SVGs in gene activity scores ($P_{\text{adj}} < 10^{-20}$), including several key transcription factors like *Zfhx4*, *Cux1*, *Cux2*, *Gpc5*. This will further support the investigation of spatially specific regulatory mechanisms.

## 5 Discussion

**Conclusion.** In this paper, we propose a principled and feasible generative model named scMRDR to integrate unpaired multi-omics single-cell data into a unified latent space. We employ $\beta$-VAE to disentangle latent embedding into modal-shared and modal-specific subspace, incorporating isometric regularization to ensure the conservation of biological information within each omics, and adversarial loss to encourage the fusion of different modalities. Masked loss are adopted to address the feature-missing issue in different modalities. Via empirical experiments and comprehensive comparison with existing methods, scMRDR exhibits an excellent performance in modality integration, bio-conservation, and batch correction, and demonstrates strong adaptability for scaling to larger, large-level datasets with more omics modalities to integrate.

**Limitations.** There are still some limitations to our approach. The regularized $\beta$-VAE introduces the trade-off between different optimization goals, reflecting in the choices of hyperparameters $\beta$, $\lambda$, and $\gamma$. In particular, the introduction of the adversarial loss, i.e., the min-max optimization objective, increases the training difficulty. Besides, we map features of all modalities to the gene level, such as aggregating scATAC-seq peak signals into gene activity scores. Although selective masking in the loss allows for partially unmatched features, it may still introduce potential information loss.

**Outlooks.** In addition to measuring various omics features within cells, emerging spatial multi-omics technologies allow us to capture the spatial coordinates of cells, further enriching cellular information. Meanwhile, perturbation sequencing enables the observation of cellular responses across omics layers to various chemical treatments and genetic perturbations (e.g., CRISPR). Integrating spatial and dynamic information across modalities under different conditions will be an important direction for future exploration and extension of scMRDR.

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

# A  Technical Appendices and Supplementary Material

## A.1  Details on experiments

The species, tissues, sample sizes, modalities, and references (the original sources) of the single-cell datasets used in the experiments are shown in Table 2. In our experiments, we set the batch size to 128, the epoch number to 200, and the dimensions of both the shared latent component ($d_u$) and modality-specific component ($d_s$) to 20. Parameters are optimized via Adam optimizer and the learning rate is started from 0.001 with a cosine annealing scheduler. Other hyperparameter settings are summarized in Table 2. 10% of the data was used as a validation set for early stopping during training based on the total loss on validation set. For scATAC-seq data, peak count signals are converted to gene-level gene activity scores using EpiScanpy. For datasets integrating two modalities, we select highly variable genes that are measured in both scRNA-seq and scATAC-seq (based on gene activity scores) as input features, and do not apply masked loss. For tri-modal integration, we use highly variable genes from scRNA-seq and scATAC-seq, as well as all genes with nonzero measurements in scProtein as input features, and apply masked loss according to the available gene list associated with each modality.

Competing methods are used with their respective default settings. Specifically, Seurat, Harmony, and scVI use gene activity scores for scATAC-seq, while all other methods operate on the original peak counts.

We run all empirical experiments on a single NVIDIA RTX 4080 GPU. The runtime of our method for 200 epochs is also reported in Table 2.

## A.2  Evaluation metrics

**F1 isolated label scores.** The optimal F1 score by optimizing the cluster assignment of the isolated label using the F1 score across Louvain clustering (resolutions 0.1–2, step 0.1). The metric is averaged across all isolated cell-type labels.

**Silhouette scores.** The global silhouette width measures (ASW) between isolated and non-isolated labels on the PCA embedding, scaled to [0, 1]. ASW measures how similar a data point is to its own cluster compared to other clusters, with higher ASW indicating more compact and well-separated clusters. For the bio-conservation, ASW was computed and averaged across cell identity labels; while for modality or batch integration, ASW was computed and averaged across batch or modality labels, and then subtracted from 1 to ensure higher scores indicate better integration.

**Kmeans NMI.** Normalized Mutual Information (NMI) measures the similarity between clustering results and known labels, accounting for label permutations. We compute NMI between KMeans clusters and ground truth labels, with scores ranging from 0 (no overlap) to 1 (perfect match).

**Kmeans ARI.** Adjusted Rand Index (ARI) evaluates clustering accuracy by considering both agreements and disagreements between predicted and true labels, adjusted for chance. We compare KMeans clusters with ground truth labels, where 0 indicates random labeling and 1 indicates perfect agreement.

**Graph LISI.** Graph LISI is an extension of the original LISI metric that is computed from neighborhood lists per node from integrated kNN graphs. Instead of relying on a fixed number of nearest neighbors, Graph LISI computes shortest-path distances on the integrated graph to consistently define neighborhoods for each cell, providing a stable diversity score even when the underlying graph has variable connectivity. The resulting scores are rescaled to a 0–1 range, where higher values indicate better batch or modality integration (iLISI) or better cell-type separation (cLISI).

**Principal component regression.** Principal Component Regression (PCR) is used to quantify batch effects by assessing how much variance in the data can be attributed to batch variables or modality differences, which is computed by multiplying the variance explained by each principal component (PC) with the $R^2$ value from regressing the batch on that PC and then summing over all PCs.

**kBET.** k-nearest neighbor Batch Effect Test (kBET) assesses data mixing by checking whether the local batch label distribution in each cell's neighborhood matches the global distribution. It reports a rejection rate across tested neighborhoods, where a lower rate indicates better batch mixing.

Table 2: Details on the single-cell datasets and experimental settings.

| Dataset | Species | Tissue | Modality | Sample size | Reference | Hyperparameters | | | | Running time |
|---|---|---|---|---|---|---|---|---|---|---|
| | | | | | | $\beta$ | $\gamma$ | $\lambda$ | Masked loss | |
| small-size two-omics integration | Human | Kidney | scRNA-seq scATAC-seq | 19,985 24,205 | [28] | 2 | 5 | 5 | No | 20min |
| large-scale two-omics integration | Mouse | Primary motor cortex | scRNA-seq scATAC-seq scRNA-seq | 69,727 54,844 30,486 | [37] | 2 | 5 | 5 | No | 120min |
| triple-omics integration | Human | Bone marrow | scATAC-seq scProtein (CITE-seq) | 10,330 18,052 | [25] | 5 | 10 | 10 | Yes | 50min |

**Graph connectivity.** Graph connectivity, ranging from 0 to 1, measures how well cells of the same identity are connected within the integrated kNN graph. For each cell type, it computes the fraction of cells in the largest connected component of that type's subgraph.

### A.3 Metric values in experiments

Unscaled metric values in all experiments are shown in Table 3. We directly used unscaled values to calculate aggregate scores.

### A.4 Ablation study and sensitivity analysis

We conducted ablation experiments and sensitivity analysis on the two-omics human kidney datasets to demonstrate the effects of applying isometric loss and adversarial learning regularization within the $\beta$-VAE framework. The results have been shown and discussed in the main text (Table 1).

To evaluate the effectiveness of the Masked loss, we also conduct ablation studies on a triple-omics dataset. The results (Table4) indicate that simply setting the missing features in the proteomics to zero without applying a loss mask leads to a significant performance drop. This highlights the importance of our masked loss strategy in effectively handling integration tasks where some omics layers (e.g., scProtein) contain substantially fewer features compared to other modalities.

### A.5 UMAP visualization of experimental results

We employ UMAP to visualize the results of multi-omics integration, where each point denotes the low-dimensional representation of an individual sample. An effective integration method should yield well-separated clusters corresponding to distinct cell types, thereby preserving the underlying biological heterogeneity. Concurrently, samples originating from different omics modalities and batches should be well-aligned in the embedding space, reflecting successful correction of modality-specific and batch-specific technical variations.

### A.6 Spatial location imputation via integration of single-cell and spatial omics

We integrated scRNA, scATAC, and spatial transcriptomics (merFISH) data in mouse primary motor cortex using scMRDR, and then interpolated the missing spatial locations in single-cell data by conducting optimal transport between the aligned latent representation $z_u$ of samples with spatial locations (merFISH) and samples without spatial information (scRNA and scATAC). We visualized the imputed spatial locations labeled by the cortex layer annotations provided by the original datasets (Fig. 12).

Table 3: Unscaled metric values in experiments

| | aggregate score | | | | bio-conservation | | | | | batch correction | | | | modality integration | | | | |
|---|---|---|---|---|---|---|---|---|---|---|---|---|---|---|---|---|---|---|
| | Overall score | Batch correct | Bio conserve | Modal integrate | IL | NMI | ARI | ASW | cLISI | ASW | iLISI | kBERT | PCR | ASW | iLISI | kBERT | GC | PCR |
| two-omics integration | | | | | | | | | | | | | | | | | | |
| Ours | 0.66 | 0.52 | 0.74 | 0.70 | 0.69 | 0.76 | 0.58 | 0.66 | 1.00 | 0.90 | 0.52 | 0.38 | 0.26 | 0.86 | 0.37 | 0.32 | 0.96 | 0.99 |
| GLUE | 0.64 | 0.42 | 0.73 | 0.74 | 0.65 | 0.77 | 0.57 | 0.67 | 1.00 | 0.90 | 0.42 | 0.28 | 0.09 | 0.85 | 0.60 | 0.34 | 0.94 | 0.99 |
| scVI | 0.56 | 0.52 | 0.65 | 0.47 | 0.59 | 0.68 | 0.43 | 0.56 | 1.00 | 0.95 | 0.47 | 0.29 | 0.37 | 0.81 | 0.00 | 0.00 | 0.71 | 0.85 |
| MaxFuse | 0.56 | 0.31 | 0.69 | 0.62 | 0.65 | 0.73 | 0.49 | 0.59 | 1.00 | 0.89 | 0.16 | 0.19 | 0.00 | 0.87 | 0.15 | 0.19 | 0.91 | 1.00 |
| Seurat | 0.54 | 0.29 | 0.75 | 0.50 | 0.68 | 0.78 | 0.61 | 0.68 | 1.00 | 0.90 | 0.06 | 0.19 | 0.00 | 0.70 | 0.00 | 0.34 | 0.46 | 0.99 |
| Pamona | 0.50 | 0.58 | 0.47 | 0.45 | 0.50 | 0.22 | 0.18 | 0.50 | 0.96 | 0.93 | 0.51 | 0.26 | 0.61 | 0.74 | 0.00 | 0.09 | 0.43 | 1.00 |
| JAMIE | 0.48 | 0.55 | 0.47 | 0.42 | 0.43 | 0.30 | 0.17 | 0.43 | 1.00 | 0.86 | 0.26 | 0.27 | 0.81 | 0.56 | 0.00 | 0.08 | 0.45 | 0.99 |
| SIMBA | 0.46 | 0.73 | 0.35 | 0.35 | 0.49 | 0.01 | 0.01 | 0.49 | 0.76 | 0.98 | 0.70 | 0.27 | 0.95 | 0.90 | 0.00 | 0.00 | 0.03 | 0.83 |
| Harmony | 0.46 | 0.34 | 0.62 | 0.37 | 0.54 | 0.60 | 0.41 | 0.54 | 1.00 | 0.89 | 0.19 | 0.27 | 0.00 | 0.56 | 0.00 | 0.06 | 0.59 | 0.64 |
| UnionCom | 0.42 | 0.41 | 0.47 | 0.36 | 0.42 | 0.43 | 0.08 | 0.44 | 0.98 | 0.81 | 0.43 | 0.39 | 0.00 | 0.38 | 0.00 | 0.00 | 0.41 | 1.00 |
| large-scale two-omics integration | | | | | | | | | | | | | | | | | | |
| Ours | 0.60 | 0.56 | 0.62 | 0.62 | 0.58 | 0.58 | 0.37 | 0.56 | 1.00 | 0.90 | 0.24 | 0.14 | 0.97 | 0.82 | 0.26 | 0.10 | 0.95 | 1.00 |
| Seurat | 0.59 | 0.53 | 0.68 | 0.53 | 0.61 | 0.69 | 0.51 | 0.61 | 1.00 | 0.86 | 0.18 | 0.09 | 0.98 | 0.77 | 0.01 | 0.07 | 0.80 | 1.00 |
| GLUE(LSI) | 0.55 | 0.55 | 0.59 | 0.49 | 0.53 | 0.55 | 0.33 | 0.53 | 0.99 | 0.90 | 0.23 | 0.09 | 0.98 | 0.52 | 0.10 | 0.11 | 0.75 | 1.00 |
| scVI | 0.54 | 0.52 | 0.57 | 0.50 | 0.53 | 0.57 | 0.26 | 0.52 | 1.00 | 0.94 | 0.12 | 0.10 | 0.93 | 0.72 | 0.00 | 0.00 | 0.82 | 0.95 |
| Harmony | 0.46 | 0.45 | 0.54 | 0.37 | 0.52 | 0.47 | 0.21 | 0.51 | 1.00 | 0.93 | 0.10 | 0.11 | 0.67 | 0.57 | 0.00 | 0.00 | 0.58 | 0.68 |
| GLUE(PCA) | 0.45 | 0.47 | 0.49 | 0.37 | 0.44 | 0.38 | 0.21 | 0.42 | 0.99 | 0.70 | 0.14 | 0.09 | 0.95 | 0.41 | 0.00 | 0.00 | 0.48 | 0.96 |
| MaxFuse | 0.43 | 0.46 | 0.41 | 0.42 | 0.42 | 0.26 | 0.01 | 0.40 | 0.97 | 0.66 | 0.13 | 0.05 | 0.99 | 0.66 | 0.00 | 0.01 | 0.45 | 1.00 |
| SIMBA | 0.41 | 0.51 | 0.35 | 0.38 | 0.49 | 0.01 | 0.00 | 0.49 | 0.73 | 0.98 | 0.12 | 0.10 | 0.85 | 0.89 | 0.00 | 0.10 | 0.07 | 0.85 |
| triple-omics integration | | | | | | | | | | | | | | | | | | |
| Ours | 0.58 | 0.39 | 0.69 | 0.61 | 0.57 | 0.68 | 0.58 | 0.61 | 1.00 | 0.83 | 0.27 | 0.16 | 0.31 | 0.83 | 0.28 | 0.06 | 0.87 | 1.00 |
| GLUE | 0.55 | 0.36 | 0.67 | 0.58 | 0.55 | 0.67 | 0.55 | 0.59 | 1.00 | 0.86 | 0.20 | 0.13 | 0.25 | 0.85 | 0.15 | 0.09 | 0.82 | 0.99 |
| scVI | 0.49 | 0.49 | 0.58 | 0.38 | 0.53 | 0.50 | 0.31 | 0.54 | 1.00 | 0.90 | 0.20 | 0.32 | 0.53 | 0.76 | 0.00 | 0.00 | 0.46 | 0.70 |
| Harmony | 0.33 | 0.29 | 0.48 | 0.18 | 0.45 | 0.35 | 0.16 | 0.44 | 1.00 | 0.76 | 0.17 | 0.11 | 0.12 | 0.51 | 0.00 | 0.00 | 0.40 | 0.00 |

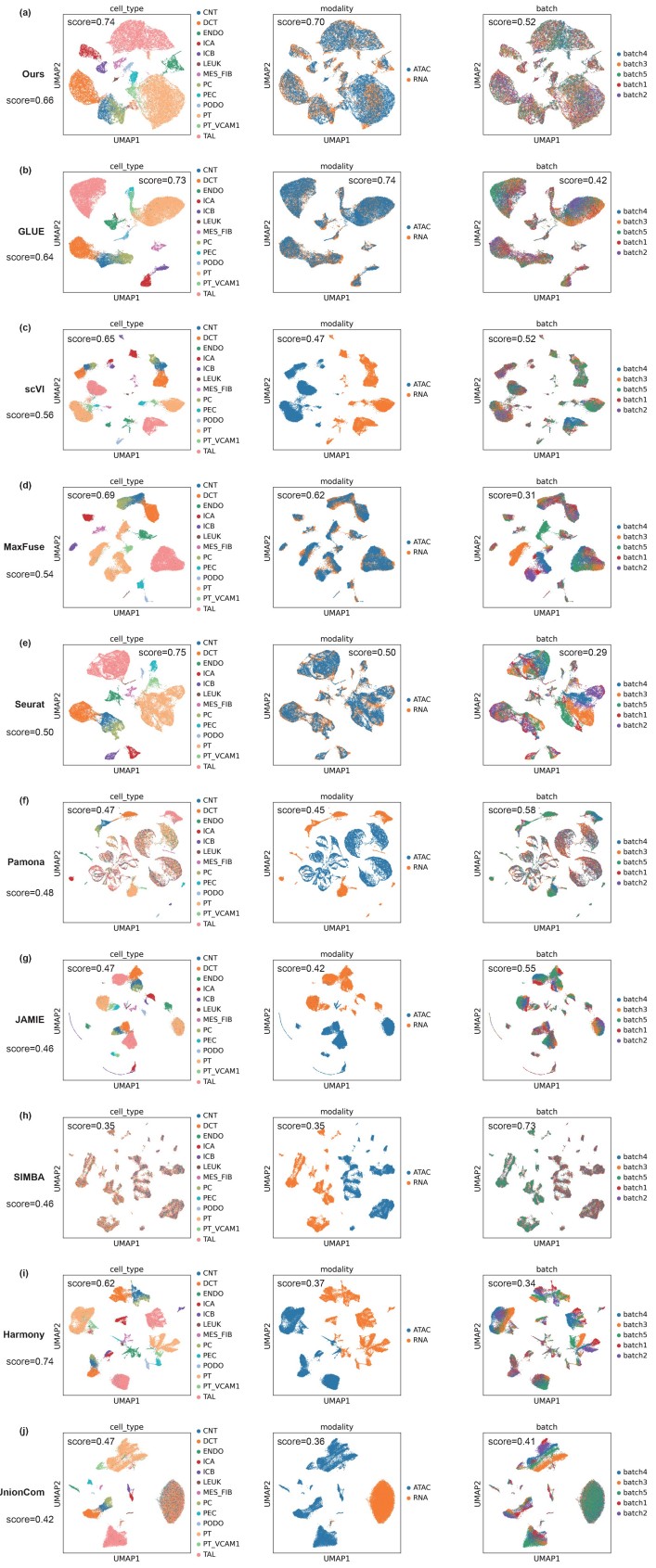

Figure 9: UMAP visualization of the unified embeddings in small-scale two-omics data integration.

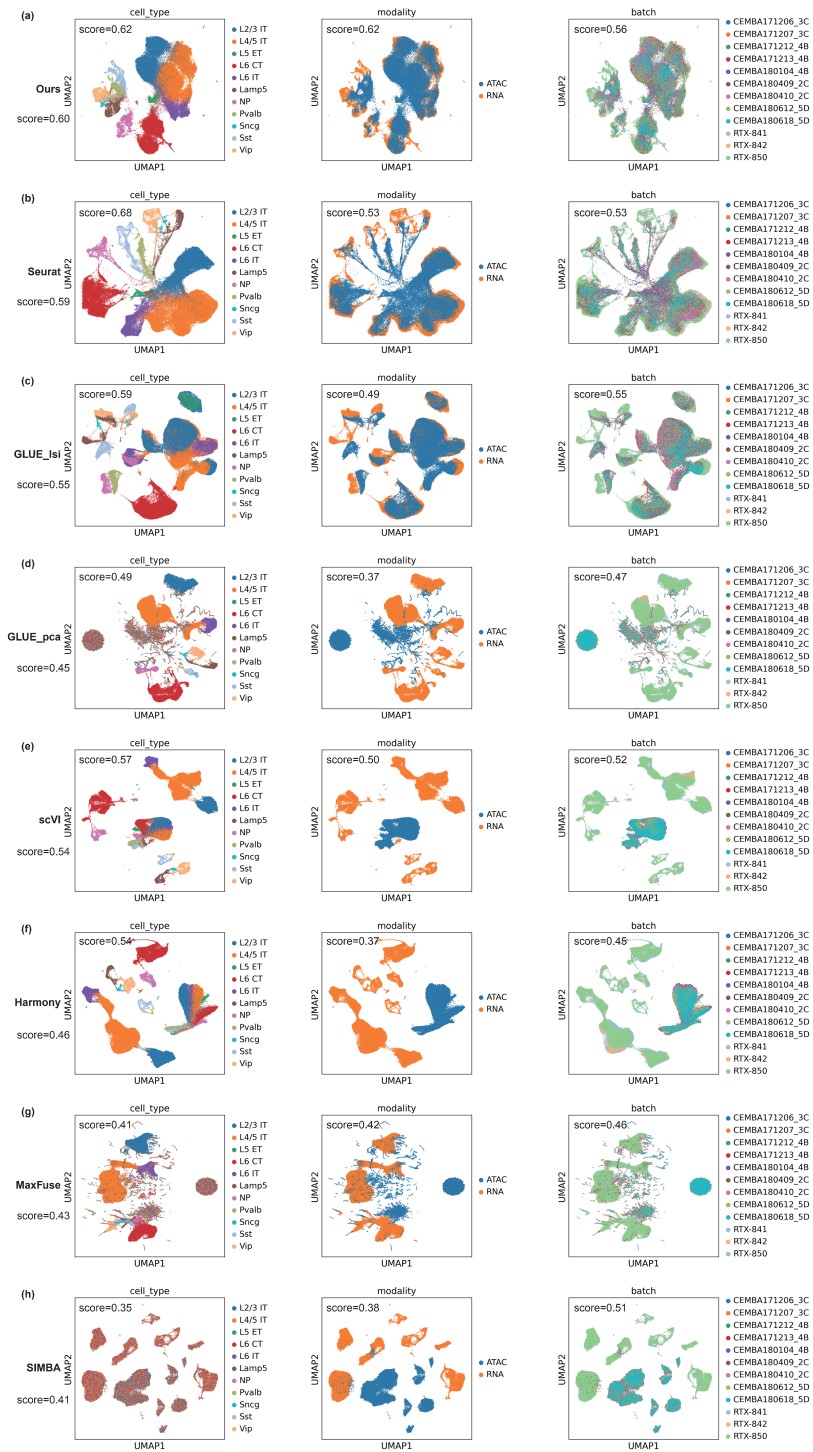

Figure 10: UMAP visualization of the unified embeddings in large-scale two-omics data integration.

Table 4: Ablation study in triple-omics data, where unscaled scIB aggregate scores are reported.

|  | Overall score | Batch correct | Bio conserve | Modal integrate |
|---|---|---|---|---|
| Ours | 0.58 | 0.39 | 0.69 | 0.61 |
| beta=1 | 0.51 | 0.39 | 0.58 | 0.52 |
| lambda=0 | 0.48 | 0.32 | 0.60 | 0.49 |
| gamma=0 | 0.53 | 0.40 | 0.58 | 0.58 |
| w/o masked loss | 0.44 | 0.33 | 0.58 | 0.37 |

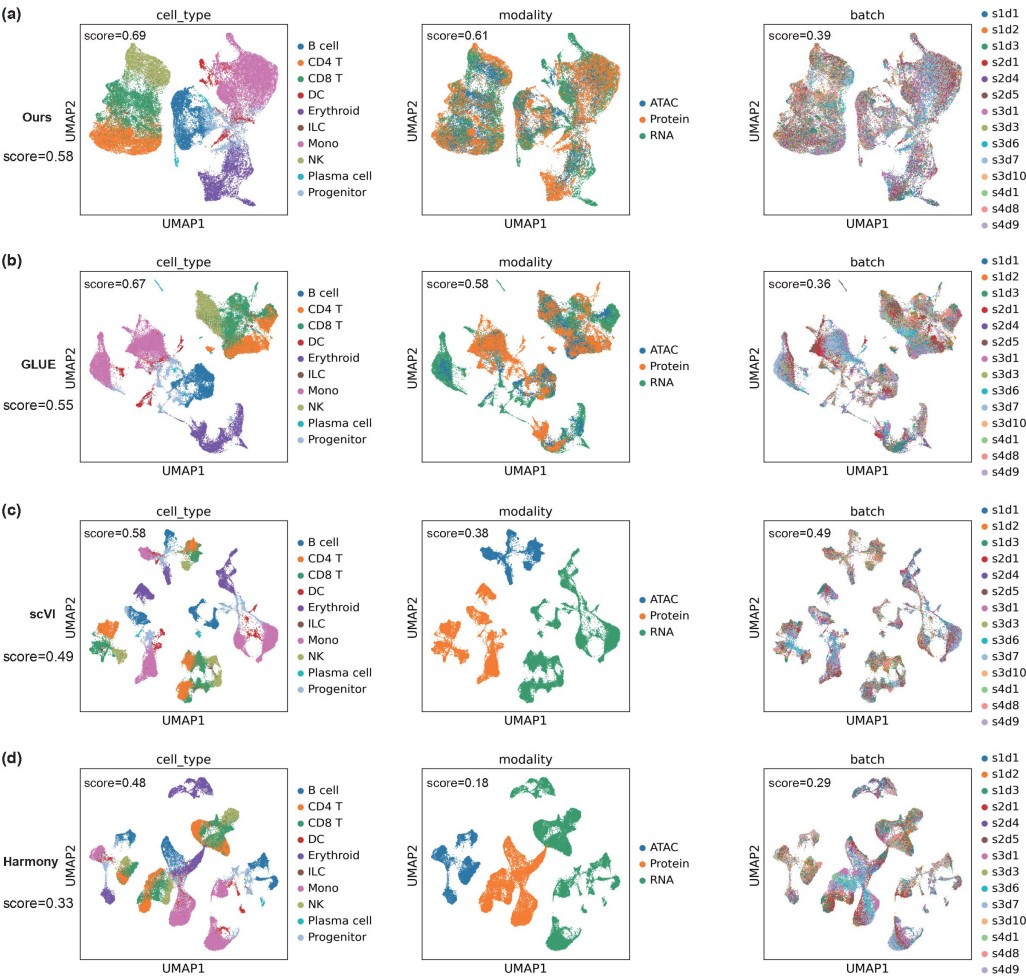

Figure 11: UMAP visualization of the unified embeddings in triple-omics data.

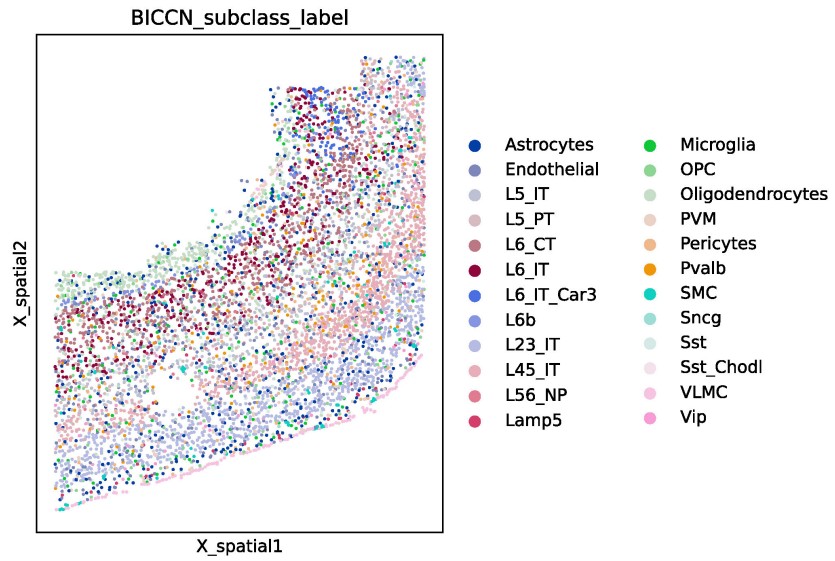

(a) Spatial plot of the merFISH data with originally measured spatial locations.

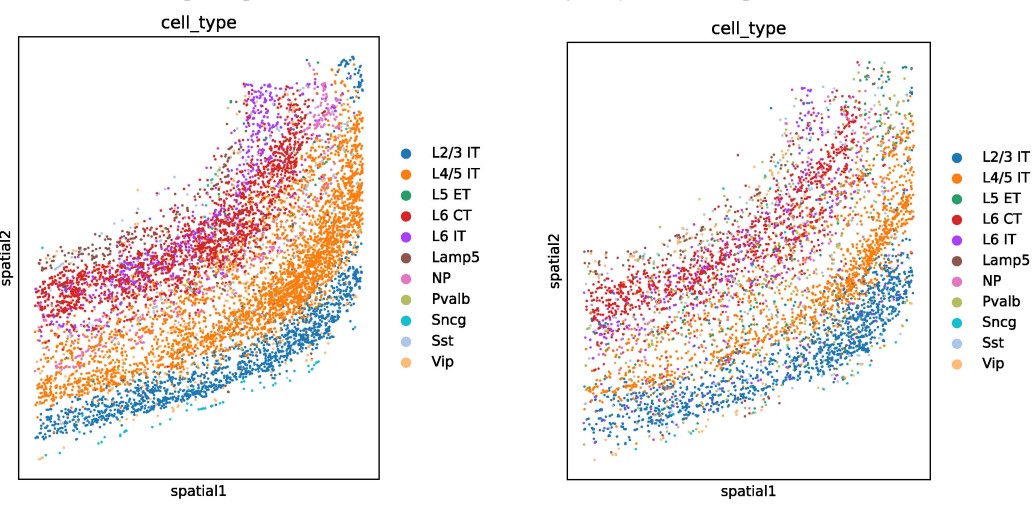

(b) Spatial plot of the scRNA data with imputed spatial locations.

(c) Spatial plot of the scATAC data with imputed spatial locations.

Figure 12: Spatial plots of merFISH and spatially imputed scRNA and scATAC data.

