# OpenReview forum: "scMRDR: A scalable and flexible framework for unpaired single-cell multi-omics data integration"
_NeurIPS.cc/2025/Conference — NeurIPS 2025 spotlight_

### Official Review · Reviewer_rgSu · 2025-07-01

**Clarity:** 3
**Significance:** 1
**Originality:** 2
**Rating:** 4
**Confidence:** 3

**Summary:**

This paper introduces scMRDR, a generative framework for unpaired single-cell multi-omics integration. The model combines several established ideas: it factorizes the latent space into modality-shared and modality-specific components via a β-VAE; aligns the shared latents using an adversarial objective; preserves biological structure using a novel isometric loss; and handles missing features through masked reconstruction. The framework is modality-agnostic, scales to large datasets, and is evaluated across three integration settings, including a triple-omics scenario and a 125k-cell atlas-scale benchmark.

**Questions:**

Clarify novelty relative to scTFBridge, scMaui, and InClust+: These methods also disentangle shared and specific latent spaces, and handle missing data through masking or PoE. Please articulate what distinguishes scMRDR beyond combining existing strategies.

Explain GLUE’s underperformance: The results shown for GLUE are much lower than expected based on its original paper and follow-ups. Was GLUE tuned properly (e.g., were feature graphs provided)? A sensitivity analysis or clarification would help.

Report raw scIB metrics: Normalized scores across models introduce instability depending on which methods are included. Please provide raw metrics or at least supplement the current tables with them.

Why exclude totalVI as a baseline? Even if it assumes paired data, totalVI is a standard method for multimodal integration and would serve as a strong point of reference.

Demonstrate downstream impact: Can you show whether scMRDR’s integrated latent improves biological tasks beyond mixing and clustering (e.g., improved classification, rare population detection, or gene regulation inference)? This would make the model’s practical benefits clearer.

**Ethical Concerns:**

["NO or VERY MINOR ethics concerns only"]

**Final Justification:**

- The rebuttal clarified key concerns, particularly around GLUE baseline, scIB metrics, and added spatial transcriptomics results.
- These additions improve the paper’s clarity and demonstrate biological utility.
- However, the main concern, limited conceptual novelty, remains largely unresolved.
- The method combines existing ideas (e.g., β-VAE, adversarial training) in an effective but incremental way.
- I agree with other reviewers that the paper is well-executed but overstates its novelty.

Updated score: 4, acknowledging improved presentation and practical value.

**Limitations:**

yes

**Quality:**

3

**Strengths And Weaknesses:**

The paper is well-written and enjoyable to read. It addresses a relevant and technically challenging problem and provides a thoughtfully engineered solution. The authors successfully integrate a combination of useful techniques - latent space separation, adversarial alignment, structure-preserving regularization, and masked losses - into a scalable, flexible framework for unpaired integration. The use of an isometric loss to preserve intra-modality structure is a novel and effective addition, and the masked loss correctly handles missing features without requiring strong assumptions or imputation. The model’s scalability is also convincingly demonstrated.

However, the overall degree of novelty is more modest than the paper suggests. While each component is justified and works well in practice, the key ideas have been explored in prior work. scTFBridge, scMaui, InClust+, and scCross all implement variations of modality-specific/shared latent decomposition, adversarial or PoE-based alignment, and masking for missing data. scMRDR improves on these by combining them more cleanly, but it is not clear whether the combination leads to fundamentally new insights or just better engineering. The use of β-VAE is a welcome inclusion, but not unprecedented. The isometric regularization stands out as the most novel component and is likely responsible for the model’s strong performance on biological conservation metrics.

The empirical results are competitive and include both large-scale and multi-modal integration, which is commendable. That said, significant concerns remain. The poor performance of GLUE is surprising and warrants clarification. Baselines such as totalVI are missing, despite being widely used and relevant for comparison. The scIB metrics (which should be cited!) should not be normalized before aggregation, as this makes them unstable to adding bad models. Finally, the evaluation focuses almost exclusively on embedding quality and integration metrics; there is no demonstration of downstream utility on real tasks (e.g., differential expression, trajectory inference, or regulatory discovery), which would help contextualize the value of improved integration.

In sum, this is an interesting contribution that addresses integration rigorously and at scale, and incorporates an architectural addition (isometric loss), however requires some revisions. The main critique is not about correctness or utility - but that the paper overclaims novelty relative to a field where many of the individual ideas have appeared in recent concurrent work, which questions the significance of this work.

---

> ### Author Rebuttal · Authors · 2025-07-31
>
> Thank you for your thoughtful review and for highlighting the strengths of our work, including the paper's clarity, the effective integration of key techniques into a scalable framework, and the novelty of beta-VAE for disentanglement and isometric loss in preserving biological structure. We also appreciate your recognition of our rigorous evaluation across multiple integration scenarios. We appreciate your feedback and believe the concerns raised—regarding novelty, baseline comparisons, and evaluation—stem from some misunderstandings that we can clearly address in our response.
>
> 1. Novelty compared to other disentanglement methods
>
> Although there have been some previous methods that decouple the latent space, they are still limited to the integration of two modalities (such as contrastive learning) or require the use of partial paired information.
> - **scTFBridge** uses a disentangled hierarchical VAE but requires **fully paired data** for training and is restricted to **dual-modality** integration (due to contrastive learning).
> - **scMaui** leverages variational product-of-experts autoencoders and adversarial learning, but also limits in **paired data**.
> - **InClust+** employs a mask-enhanced VAE with stacked modules to integrate multimodal data. Although it can achieve integration of unpaired multi-omics data, this still requires **separate processing between each pair of datasets**.
>
> In contrast, the innovation of **our method** lies in：
> - Instead of distinguishing data from different modalities at the input stage, we treat them equally as a single sample, and directly leveraging beta-VAE for disentanglement, which allows us to **avoid limitations in scalability and flexibility imposed by designs such as partial paired information supervision, stacked encoders, and cross-modal contrastive learning**, and therefore, it is conveniently adaptable to the integration of **completely unpaired data across multiple omics**.
> - As you pointed out, to ensure that the disentangled representations retain meaningful biological heterogeneity, we first and innovatively introduced an **isometric regularization**. Adversarial loss and mask loss have indeed appeared in previous articles, but their combination in this paper is not merely a simple engineering trick. Instead, it is an integral part of our novel framework, which is designed to accommodate the concatenation of all modal data and the unified use of a single beta-VAE to learn disentangled and meaningful representations, rooted in the theory of disentangled representation learning.
> - Theoratically, without additional constraints, such a disentangled subspaces are unidentifiable (i.e., not unique) [1]. We leverage this unidentifiability and, by imposing isometric and adversarial regularization, constrain the modality-shared subspace to be the one that preserves the maximum sample structure information from the original latent space while aligning different modalities, thereby endowing the unified embedding with biological significance.
>
> 2. GLUE's performance in section 4.3
>
> The comparative method GLUE demonstrates good performance in Sections 4.2 and 4.4, but is significantly lower in Section 4.3. This is because:
> * **Scaled metrics**: We initially displayed all metrics after min-max scaling (scIB's default setting), which resulted in the lowest values being scaled to 0. In fact, the raw metric values are not abnormally low (Table 2 in Appendix). We recognize that this presentation method could lead to widespread misunderstandings and will present the **original values** of each metric in the revised main text.
> * **Preprocessing methods**: GLUE recommends using LSI for the preprocessing of scATAC, but the function `scglue.data.lsi` provided by the GLUE fails to deal with the scale of 54,844 cells (keep returning NaN values). Therefore, in the initial version of the paper, we substituted PCA (the preprocessing method for scRNA in GLUE), which resulted in a significant drop in model performance. We now used the LSI method for large-scale sparse data in sklearn `TruncatedSVD`, and there's a significant improvement in GLUE's performance, though it still falls short of our method. This highlights another advantage of our method: it **does not rely on any preprocessing methods**.
> * Besides, as you suggested, we conducted **sensitivity analysis** for GLUE: we compare the performance of GLUE under 8 settings: preprocessing methods (LSI / PCA) x modeling layers (Negative Binomial for raw counts / Normal for normalized expressions) x whether to use highly variables genes. **All configurations of GLUE performs worse than our method**.
> ### Sensitivity analysis of GLUE in large-scale two-omics integration
> ||overall score|batch correct|bio-conserve|modal integrate|
> |----------------------|---------|---------------|--------------|-----------------|
> |Ours|0.60|0.56|0.62|0.62|
> |GLUE(LSI+NB+hvg)|0.55|0.55|0.59|0.49|
> |GLUE(LSI+NB)|0.53|0.54|0.58|0.46|
> |GLUE(LSI+Normal+hvg)|0.51|0.51|0.54|0.46|
> |GLUE(LSI+Normal)|0.48|0.51|0.48|0.44|
> |GLUE(PCA+Normal)|0.45|0.48|0.47|0.39|
> |GLUE(PCA+NB+hvg)|0.45|0.48|0.48|0.37|
> |GLUE(PCA+NB)|0.44|0.47|0.47|0.37|
> |GLUE(PCA+Normal+hvg)|0.43|0.46|0.42|0.39|
>
> 3. raw scIB metrics
>
> We apologize for the widespread misunderstanding caused by our previous presentation.
> * For convenience in calculating the summary metrics, we used **scIB's default settings** (we cited scIB in line 197-198), which involved **min-max scaling** each metric to 0-1 before summing them up, and we directly visualized the **scaled scores** in the main text (We explained it in the captions of each figure)
> * we attached the original unscaled metrics in the appendix (Table 2). We also mentioned that in the captions of Figures 4, 5, and 7.
> * We recognize that this approach has led to widespread misunderstandings, particularly as scaling can further lower the values of some methods. Therefore, we are now updating to report **unscaled metric values** and aggregated scores based on unscaled values in the revised main text. (**Please see the raw values in our response to Reviewer XHw7 due to space limitations**)
>
> 4. totalVI as a baseline
>
> As you pointed out, totalVI is an integration method for paired data (each cell are simultaneously measured in multiple omics), which is fundamentally different from the problem we want to address - integrating cells with single-modality measurements from different datasets (unpaired/diagonal integration). Therefore, we excluded it.
>
> In **paired integration**, the integration method utilizes multi-omics features to obtain a latent embedding for each cell. Each multi-modal cell has only one embedding. In contrast, **unpaired integration** aims to align cells measured across various modalities from different datasets. Each cell in each modality will have its own embedding, required to preserve biological heterogeneity information in each modality while aligning the data across different modalities.
>
> Relevant benchmarks [2-3] all distinguish the two scenarios and corresponding methods. Even assuming unpaired data as paired, totalVI cannot handle it (due to unequivalent sample sizes across modalities and it generating a single embedding for each pair).
>
> 5. downstream impact
>   - We added an additional downstream analysis: **spatial location imputation** of scRNA and scATAC.
>   - We integrated scRNA, scATAC, and spatial transcriptomics (merFISH) [4] data using our method, and then used the aligned latent representation to **interpolate the missing spatial locations in single-cell data** by optimal transport. Visualization shows that this interpolation performs well, where inferred locations of cells align well with the provided cortex layers annotations. We apologize that we are not allowed to include the figures here due to the rebuttal guidelines, but they will be presented in the revised version of the manuscript.
>   - Detecting spatial variable genes (SVGs): Due to low coverage of merFISH (only **254** genes), only **103** genes are detected as SVGs (SPARKX $P_{adj}<1e-20$). We leveraged the spatially interpolated scRNA data (**26069** genes) and **4063** SVGs (SPARKX $P_{adj}<1e-20$) are detected. We replicated **85** out of 101 SVGs detectable by merFISH (such as *Lamb5*, *Calb1*, *Cux2*, etc.), and also revealed new SVGs (such as *Sema5b*, *Gpr88*, *Zfhx4*, etc.). These findings are also supported by some previous studies [5-6].
>   - Similarly, using **scATAC with imputed spatial locations**, we identified **386** SVGs on gene activity scores, including several key transcription factors (such as *Zfhx4*, *Cux1*, *Cux2*, *Foxp1*, etc.). This will further support the investigation of spatially specific regulatory mechanisms.
>   - **This indicates that our integration method preserves the unique biological information of each cell and can be used to further impute missing spatial location information, thereby enabling more biological discoveries.**
>
> Reference:
>
> [1] Khemakhem, et al. Variational autoencoders and nonlinear ica: A unifying framework. International conference on artificial intelligence and statistics (2020)
>
> [2] Fu, S. et al. Benchmarking single-cell multi-modal data integrations. Nat Methods (2025)
>
> [3] Chuxi X. et al. Benchmarking multi-omics integration algorithms across single-cell RNA and ATAC data, Brief in Bioinfo (2024)
>
> [4] Zhang, M. et al. Spatially resolved cell atlas of the mouse primary motor cortex by MERFISH. Nature (2021).
>
> [5] Stickels, R.R. et al. Highly sensitive spatial transcriptomics at near-cellular resolution with Slide-seqV2. Nat Biotechnology (2021).
>
> [6] Liau, E.S. et al. Single-cell transcriptomic analysis reveals diversity within mammalian spinal motor neurons. Nat Commun (2023).

---

> > ### Comment · Reviewer_rgSu · 2025-08-04
> >
> > Thank you for the detailed rebuttal, which clarifies many details and improves transparency around evaluation, particularly regarding GLUE's performance and scIB metrics. I appreciate the additional results and downstream analyses on spatial transcriptomics, which strengthen the biological relevance of the method.
> >
> > That said, my central concern remains: the method combines known concepts in a way that seems effective (though I share reviewer co9u's concern about the evaluation), but not conceptually novel. While the authors emphasise distinctions in pairing assumptions and architectural details, these feel incremental relative to the mentioned prior work, which also address representation disentanglement in a very similar manner. The isometric regularisation is a nice addition, though its impact and theoretical motivation could be better isolated. I agree with co9u thtat the novelty claim is somewhat overstated. The improvements in evaluation clarity and downstream results are appreciated, but do not fully address this core concern.
> >
> > Given the stronger positioning of the work after the rebuttal, I would consider increasing my score slightly (from 3 to 4), recognising the paper's value as a well-engineered system.

---

> ### Author Response · Authors · 2025-08-04
>
> Thank you for your thoughtful feedback and for recognizing the improvements in evaluation clarity and the additional downstream analyses on spatial transcriptomics.
>
> We would like to emphasize that the significance of our work lies in providing a convenient framework for integrating **non-paired multi-omics data without relying on specific pairing information or conventional pairwise transfer**. To achieve this, we established a framework with **a unified encoder treating observation in different omics equally as a single sample** (instead of stacked specific encoders) for disentangled learning for the first time. Further, we proposed a joint optimization goal, combining our novel use of isometric regularization with existing techniques, enabling our flexible framework effective. Experiments demonstrate its scalability in large sample size and over 2 omics. We agree your valuable suggestion and will clarify the theoretical motivation for regularized disentangled representations better in the revised manuscript.

---

### Official Review · Reviewer_co9u · 2025-07-02

**Clarity:** 3
**Significance:** 3
**Originality:** 2
**Rating:** 4
**Confidence:** 4

**Summary:**

This paper proposes scMRDR, a generative framework based on $\beta$-VAE for integrating unpaired single-cell multi-omics data. The method disentangles latent representations into modality-shared and modality-specific components, and incorporates isometric regularization, adversarial training, and masked reconstruction loss. The authors claim superior scalability and flexibility compared to existing methods, demonstrating improved performance on two-omics and triple-omics integration tasks.

**Questions:**

Why does GLUE perform well in triple-omic integration (Figure 6) but performs poorly in two-omics integration (Figure 5)?

**Ethical Concerns:**

["NO or VERY MINOR ethics concerns only"]

**Final Justification:**

- The rebuttal has clarified my concerns, including the GLUE benchmark and significance of the method
- The method was well motivated to solve the unpaired multi-omic integration task
- As shared among reviewers, the novelty is limited to the combination of beta-VAE and the use of isometric loss and adversarial training

**Limitations:**

yes

**Quality:**

2

**Strengths And Weaknesses:**

- Strengths
  1. The unpaired omics integration problem  is well motivated. scMRDR improves the multi-omic integration in a scalable way.
  2. scMRDR achieves very competitive results across multiple metrics, comparable to strong baselines.
- Weaknesses
  1. It is inappropriate to call a dataset of 69,727 + 54,844 cells "atlas"-level. I understand this Section 4.3 shows the scalbility of the methods, and many other baselines could not process this number of cells, but it falls short of true atlas scales (at least at millions). This section overstates the difficulty of the scalability benchmark, ignoring the more complex multi-assay scenarios that current atlas efforts involve. Similar methods, e.g. MultiVI [1], have multi scRNA-seq assay benchmark.
  2. Although ablation shows each loss component is essential to model performance, the loss weights are non-trivial and not systematically characterized. The selection of hyperparameters is finetuned for tasks (Appendix A.2, page 13).

[1] Ashuach, Tal, et al. "MultiVI: deep generative model for the integration of multimodal data." Nature Methods 20.8 (2023): 1222-1231.

---

> ### Author Rebuttal · Authors · 2025-07-31
>
> Thank you for recognizing the strong motivation of our work on unpaired omics integration and for highlighting scMRDR's scalable approach and its very competitive performance across multiple metrics. We appreciate your acknowledgment of the method's effectiveness in both two-omics and triple-omics integration tasks. We appreciate your insightful questions and we believe the concerns have been well clarified below.
> 1. On the "atlas" scale
>
> * We recognize that our experiment in section 4.3 (69,727 + 54,844 cells) does not truly qualify as atlas-level, which may require millions of cells; therefore, we will change the term "atlas-scale" to "large-scale" in the final version of the paper.
> * Meanwhile, we believe this does not diminish the **significance** of our method, because
>   * (a) as you pointed out, we demonstrated the **scalability** of our method through the integration of cells at the 100,000-cell level, while many comparative methods fail to achieve this;
>   * (b) As shown in the latest benchmark article published on this month [1], many of the newly published methods over the past two years can still only perform integration on paried datasets (paired integration) or partial paired dataset (mosaic integration), while methods supporting **complete unpaired data integration** are still scarce (14 out of 40), especially for integrating completely unpaired data across **more than two omics**, which highlights the significance of our work for its capacities in **completely unpaired (diagonal) integration with over two omics**.
> 2. On the hyperparameters
>
> Based on the original ablation experiments, we conducted experiments with a range of **different hyperparameter combinations** on the regular-scale two-omics dataset.
> * The performance is consistently better than the ablation results when removing any individual component, and the worst model is the baseline without any regularization. While different hyperparameter combinations have a certain impact on performance, including trade-offs among various losses and the influence of randomness (isometry is more essential to bio-conservation while adversary more vital to modality alignment, but simply increasing the value of a certain parameter will not necessarily enhance the model's performance in a specific aspect), the results are **generally robust** to the choice of hyperparameters.
> * Our design is not by chance. Theoratically, without additional constraints, such a disentangled subspaces are unidentifiable (i.e., not unique) [2]. We leverage this unidentifiability and, by imposing isometric and adversarial regularization, constrain the modality-shared subspace to be the one that preserves the maximum sample structure information from the original latent space while aligning different modalities, thereby endowing the unified embedding with biological significance.
> ### Performance under different hyperparameter configurations
> |beta|gamma|lambda|overall score|batch correct|bio-conserve|modal integrate|notes|
> |------|-------|--------|-------|---------------|--------------|-----------------|----------------|
> |2|5|5|0.66|0.52|0.74|0.70||
> |2|7|5|0.65|0.54|0.69|0.72||
> |2|5|3|0.65|0.55|0.69|0.71||
> |2|2|2|0.65|0.50|0.72|0.70||
> |2|5|7|0.65|0.49|0.70|0.73||
> |2|3|5|0.65|0.47|0.71|0.74||
> |2|1|5|0.64|0.49|0.68|0.75||
> |3|5|5|0.64|0.48|0.71|0.71||
> |4|5|5|0.64|0.47|0.71|0.72||
> |2|10|5|0.64|0.52|0.68|0.71||
> |1|5|5|0.62|0.50|0.67|0.68|beta=1|
> |2|5|0|0.61|0.53|0.69|0.60|no adversarial|
> |2|0|5|0.61|0.51|0.60|0.72|no isometric|
> |1|0|0|0.59|0.48|0.63|0.66|baseline|
>
> 3. On the performance of scGLUE in section 4.3
>
> We apologize for any misunderstandings caused by the previous presentation. In Section 4.3, the reasons for the lower metric values of GLUE include:
> * **Scaled metrics**: We initially displayed all metrics after min-max scaling (scIB's default setting), which resulted in the lowest values being scaled to 0. In fact, the raw metric values are not abnormally low (Table 2 in Appendix). We recognize that this presentation method could lead to widespread misunderstandings and will present the **original values** of each metric in the revised main text.
> * **Preprocessing methods**: GLUE recommends using LSI for the preprocessing of scATAC, but the function `scglue.data.lsi` provided by the GLUE fails to deal with the scale of 54,844 cells (keep returning NaN values). Therefore, in the initial version of the paper, we substituted PCA (the preprocessing method for scRNA in GLUE), which resulted in a significant drop in model performance. We now used the LSI method for large-scale sparse data in sklearn `TruncatedSVD`, and there's a significant improvement in GLUE's performance, though it still falls short of our method. This highlights another advantage of our method: it **does not rely on any preprocessing methods**.
> * We conducted **sensitivity analysis** for GLUE: we compare the performance of GLUE under 8 settings: preprocessing methods (LSI / PCA) x modeling layers (Negative Binomial for raw counts / normal for normalized expressions) x whether to use highly variables genes. **All configurations of GLUE performs worse than our method**.
> ### Performance of GLUE in large-scale two-omics integration
> ||overall score|batch correct|bio-conserve|modal integrate|
> |----------------------|---------|---------------|--------------|-----------------|
> |Ours|0.60|0.56|0.62|0.62|
> |GLUE(LSI+NB+hvg)|0.55|0.55|0.59|0.49|
> |GLUE(LSI+NB)|0.53|0.54|0.58|0.46|
> |GLUE(LSI+Normal+hvg)|0.51|0.51|0.54|0.46|
> |GLUE(LSI+Normal)|0.48|0.51|0.48|0.44|
> |GLUE(PCA+Normal)|0.45|0.48|0.47|0.39|
> |GLUE(PCA+NB+hvg)|0.45|0.48|0.48|0.37|
> |GLUE(PCA+NB)|0.44|0.47|0.47|0.37|
> |GLUE(PCA+Normal+hvg)|0.43|0.46|0.42|0.39|
>
> Reference
>
> [1] Fu, S. et al. Benchmarking single-cell multi-modal data integrations. Nat Methods (2025)
>
> [2] Khemakhem, et al. Variational autoencoders and nonlinear ica: A unifying framework. International conference on artificial intelligence and statistics (2020)

---

> > ### Comment · Area_Chair_tUu7 · 2025-08-04
> >
> > Dear Reviewer co9u,
> >
> > The concerns you raised regarding novelty have also been noted by other reviewers (e.g., Reviewer rgSu). However, their overall evaluation is more positive, particularly after considering the authors' response.
> >
> > Could you please take a moment to review the other reviewers' comments and the authors' rebuttal? Kindly update your review and score, or clarify if your concerns remain unresolved.

---

> > ### Comment · Reviewer_co9u · 2025-08-05
> >
> > Thanks for the detailed response on the significance of the method and clarification on scGLUE benchmark! My concerns are well addressed. It would be great to present the raw values in the main text as the authors promised in their rebuttal. I've raised my score accordingly.

---

> ### Author Response · Authors · 2025-08-05
>
> Thank you for your valuable feedback and for recognizing our work. As promised, we will present the unscaled metrics in the final version of the manuscript.

---

### Official Review · Reviewer_XHw7 · 2025-07-03

**Clarity:** 3
**Significance:** 3
**Originality:** 3
**Rating:** 5
**Confidence:** 4

**Summary:**

This paper proposes an algorithm to enable multi-omics integrations on unpaired data. The authors design a disentangled VAE using the framework of $\beta$-VAE and add new regularization terms to enable the model achieves common latent space. The authors examined their model using several multi-omics datasets.

**Questions:**

1. The latent embeddings are split into two parts, shared and exclusive. Which one is used to produce UMAP and further analysis?
2. In Figure 4, for the PCR comparison of batch correction, the result of the proposed method is 0. Is it a typo? If not, what issue causes this abnormal result?
3. I would like to see whether the disentangled features are able to capture the meaningful representation.
4. In Figure 6, does the cell type label align between RNA-seq and ATAC-seq?

**Ethical Concerns:**

["NO or VERY MINOR ethics concerns only"]

**Final Justification:**

I appreciate authors' effort on providing detailed benchmarks in a short time, and I think this rebuttal has addressed all my concerns. I will raise my points accordingly. Please make sure to include these additional benchmark in the future version.

**Limitations:**

See my questions

**Quality:**

3

**Strengths And Weaknesses:**

Strengths:
1. Clear writing. I really enjoy the clear writing which has a very readable manuscript.
2. The integration of unpaired multi-omics data is an important topic.
3. The proposed method is clear and achieves a good performance compared to baselines.

Weaknesses:
1. Some results in the experiment section needs further clarifications.
2. The compared baselines are a little bit outdated. The latest baseline is GLUE, which was published three years ago.

---

> ### Author Rebuttal · Authors · 2025-07-31
>
> Thank you for your positive feedback on the clarity of our manuscript, the significance of the problem, and the performance of our method. We appreciate your insightful questions and we believe the concerns have been well clarified below.
> 1. Explantions on experimental results
>
> * We previously visualized min-max scaled metrics in Fig.4,5,7: For convenience in calculating the summary metrics, we used **scIB's default settings** [1], which involved **min-max scaling** each metric to 0-1 before getting aggregate scores, and directly visualized the **scaled scores** in the main text (explained in the captions of each figure)
> * we attached the original unscaled metrics in the appendix (Table 2). We also mentioned that in the captions of Figures 4, 5, and 7.
> * We recognize that it has led to widespread misunderstandings. Therefore, we will turn to present **unscaled metric values** and aggregated scores based on unscaled values (see tables below)
> 2. Comparison with recently published methods
>
> We referenced the newest benchmark paper published in Nature Methods this month [2], and added comparison with two new methods for unpaired multi-omics integration: **MaxFuse** [3] and **SIMBA** [4] (see tables below). Besides,
> * in this latest benchmark, GLUE remains the best-performing method among all compared methods for regular-scale unpaired integration of scRNA and scATAC, indicating that our original benchmark is still meaningful;
> * many of the newly published methods discussed in this benchmark can still only perform integration on paried datasets (paired integration) or partial paired dataset (mosaic integration), while methods supporting **complete unpaired data integration** are still scarce (14 out of 40), especially for integrating completely unpaired data across **more than two omics**. This further highlights the significance of our work.
> ### performance on two-omics integration
> ||aggregate score||||bio-conservation|||||batch correction||||modality integration|||||
> |----------|:---------------:|:-------------:|:------------:|:---------------:|:----------------:|:-----:|:-----:|:-----:|:-----:|:----------------:|:-----:|:-----:|:-----:|:--------------------:|:-----:|:-----:|:-----:|:-----:|
> ||overall|batch correct|bio conserve|modal integrate|IL|NMI|ARI|ASW|cLISI|ASW|iLISI|kBERT|PCR|ASW|iLISI|kBERT|GC|PCR|
> |Ours|0.66|0.52|0.74|0.70|0.69|0.76|0.58|0.66|1.00|0.90|0.52|0.38|0.26|0.86|0.37|0.32|0.96|0.99|
> |GLUE|0.64|0.42|0.73|0.74|0.65|0.77|0.57|0.67|1.00|0.90|0.42|0.28|0.09|0.85|0.60|0.34|0.94|0.99|
> |scVI|0.56|0.52|0.65|0.47|0.59|0.68|0.43|0.56|1.00|0.95|0.47|0.29|0.37|0.81|0.00|0.00|0.71|0.85|
> |MaxFuse|0.56|0.31|0.69|0.62|0.65|0.73|0.49|0.59|1.00|0.89|0.16|0.19|0.00|0.87|0.15|0.19|0.91|1.00|
> |Seurat|0.54|0.29|0.75|0.50|0.68|0.78|0.61|0.68|1.00|0.90|0.06|0.19|0.00|0.70|0.00|0.34|0.46|0.99|
> |Pamona|0.50|0.58|0.47|0.45|0.50|0.22|0.18|0.50|0.96|0.93|0.51|0.26|0.61|0.74|0.00|0.09|0.43|1.00|
> |JAMIE|0.48|0.55|0.47|0.42|0.43|0.30|0.17|0.43|1.00|0.86|0.26|0.27|0.81|0.56|0.00|0.08|0.45|0.99|
> |SIMBA|0.46|0.73|0.35|0.35|0.49|0.01|0.01|0.49|0.76|0.98|0.70|0.27|0.95|0.90|0.00|0.00|0.03|0.83|
> |Harmony|0.46|0.34|0.62|0.37|0.54|0.60|0.41|0.54|1.00|0.89|0.19|0.27|0.00|0.56|0.00|0.06|0.59|0.64|
> |UnionCom|0.42|0.41|0.47|0.36|0.42|0.43|0.08|0.44|0.98|0.81|0.43|0.39|0.00|0.38|0.00|0.00|0.41|1.00|
> ### Performance on large-scale two-omics integration
> ||aggregate score||||bio-conservation|||||batch correction||||modality integration|||||
> |-----------|:---------------:|:-------------:|:------------:|:---------------:|:----------------:|:-----:|:-----:|:-----:|:-----:|:----------------:|:-----:|:-----:|:-----:|:--------------------:|:-----:|:-----:|:-----:|:-----:|
> ||overall|batch correct|bio conserve|modal integrate|IL|NMI|ARI|ASW|cLISI|ASW|iLISI|kBERT|PCR|ASW|iLISI|kBERT|GC|PCR|
> |Ours|0.60|0.56|0.62|0.62|0.58|0.58|0.37|0.56|1.00|0.90|0.24|0.14|0.97|0.82|0.26|0.10|0.95|1.00|
> |Seurat|0.59|0.53|0.68|0.53|0.61|0.69|0.51|0.61|1.00|0.86|0.18|0.09|0.98|0.77|0.01|0.07|0.80|1.00|
> |GLUE(LSI)|0.55|0.55|0.59|0.49|0.53|0.55|0.33|0.53|0.99|0.90|0.23|0.09|0.98|0.52|0.10|0.11|0.75|1.00|
> |scVI|0.54|0.52|0.57|0.50|0.53|0.57|0.26|0.52|1.00|0.94|0.12|0.10|0.93|0.72|0.00|0.00|0.82|0.95|
> |Harmony|0.46|0.45|0.54|0.37|0.52|0.47|0.21|0.51|1.00|0.93|0.10|0.11|0.67|0.57|0.00|0.00|0.58|0.68|
> |GLUE(PCA)|0.45|0.47|0.49|0.37|0.44|0.38|0.21|0.42|0.99|0.70|0.14|0.09|0.95|0.41|0.00|0.00|0.48|0.96|
> |MaxFuse|0.43|0.46|0.41|0.42|0.42|0.26|0.01|0.40|0.97|0.66|0.13|0.05|0.99|0.66|0.00|0.01|0.45|1.00|
> |SIMBA|0.41|0.51|0.35|0.38|0.49|0.01|0.00|0.49|0.73|0.98|0.12|0.10|0.85|0.89|0.00|0.10|0.07|0.85|
> ### Performance on triple-omics integration
> ||aggregate score||||bio-conservation|||||batch correction||||modality integration|||||
> |---------|:---------------:|:-------------:|:------------:|:---------------:|:----------------:|:-----:|:-----:|:-----:|:-----:|:----------------:|:-----:|:-----:|:-----:|:--------------------:|:-----:|:-----:|:-----:|:-----:|
> ||overall|batch correct|bio conserve|modal integrate|IL|NMI|ARI|ASW|cLISI|ASW|iLISI|kBERT|PCR|ASW|iLISI|kBERT|GC|PCR|
> |Ours|0.58|0.39|0.69|0.61|0.57|0.68|0.58|0.61|1.00|0.83|0.27|0.16|0.31|0.83|0.28|0.06|0.87|1.00|
> |GLUE|0.55|0.36|0.67|0.58|0.55|0.67|0.55|0.59|1.00|0.86|0.20|0.13|0.25|0.85|0.15|0.09|0.82|0.99|
> |scVI|0.49|0.49|0.58|0.38|0.53|0.50|0.31|0.54|1.00|0.90|0.20|0.32|0.53|0.76|0.00|0.00|0.46|0.70|
> |Harmony|0.33|0.29|0.48|0.18|0.45|0.35|0.16|0.44|1.00|0.76|0.17|0.11|0.12|0.51|0.00|0.00|0.40|0.00|
>
> 3. Clarifications of questions
>
> * We used the **modality-shared latent embeddings** for UMAP visualization and subsequent analysis.
> * **abnormal metrics values**: As explained before, we initially displayed all metrics after **min-max scaling** (default setting of scIB package), which scales the lowest values to 0. Though we explained it in the captions of each figure, we recognize that it leads to widespread misunderstandings, and we will present the **raw values** in the revised main text.
> * We effectively capture **biologically meaningful representations**:
>   - Our modality integration method demonstrates excellent **bio-conservation**: despite an **unsupervised** manner, different cell types are separated into distinct clusters, while the same cell type across different modalities cluster together in the UMAP visulaization.
>   - Additional downstream analysis: We integrated scRNA, scATAC, and spatial transcriptomics (merFISH) using our method, and then used the aligned latent representation to **interpolate the missing spatial locations in single-cell data** by optimal transport. Visualization shows that this interpolation performs well, where inferred locations of cells align well with the provided cortex layers annotations. We apologize that we are not allowed to include the figures here due to the rebuttal guidelines, but they will be presented in the revised version of the manuscript.
>   - Detecting spatial variable genes (SVGs): Due to the low coverage of merFISH (only **254** genes), only **103** genes are detected as SVGs (SPARKX $P_{adj}<1e-20$). We leveraged the spatially interpolated scRNA data (**26069** genes) and **4063** SVGs (SPARKX $P_{adj}<1e-20$) are detected. We replicated **85** out of 101 SVGs detectable by merFISH (like *Lamb5*, *Calb1*, *Cux2*, etc.), and also revealed new SVGs (like *Sema5b*, *Gpr88*, *Zfhx4*, etc.), which are supported by some pervious studies [5-6].
>   - Similarly, using **scATAC with imputed spatial locations**, we identified **386** SVGs in gene activity scores, including several key transcription factors like *Zfhx4*, *Cux1*, *Cux2*, *Foxp1*, etc. This will further support the investigation of spatially specific regulatory mechanisms.
>   - **This indicates that our integration method preserves the unique biological information of each cell and can be used to further impute missing spatial location information, thereby enabling more biological discoveries.**
> * Cell type labels in different omics has been **aligned** between different modalities. It should be emphasized that we did not use these labels during training (**completely unsupervised**); cell type labels were only used for evaluation.
>
> Reference
>
> [1] Luecken, M.D. et al. Benchmarking atlas-level data integration in single-cell genomics. Nat Methods (2022)
>
> [2] Fu, S. et al. Benchmarking single-cell multi-modal data integrations. Nat Methods (2025)
>
> [3] Chen, S. et al. Integration of spatial and single-cell data across modalities with weakly linked features. Nat Biotechnol (2024)
>
> [4] Chen, H. et al. SIMBA: single-cell embedding along with features. Nat Methods (2024)
>
> [5] Stickels, R.R. et al. Highly sensitive spatial transcriptomics at near-cellular resolution with Slide-seqV2. Nat Biotechnology (2021).
>
> [6] Liau, E.S. et al. Single-cell transcriptomic analysis reveals diversity within mammalian spinal motor neurons. Nat Commun (2023).

---

> > ### Comment · Area_Chair_tUu7 · 2025-08-04
> > **Follow-up on Paper 4013**
> >
> > Dear Reviewer XHw7,
> >
> > The authors have provided additional clarifications and included comparisons to newer baselines in response to your comments.
> >
> > Could you please confirm whether their response addresses the weaknesses you raised? If so, kindly update your review and score. If not, please clarify any remaining concerns.

---

> > ### Author Response · Authors · 2025-08-06
> >
> > Dear reviewer XHw7,
> >
> > We sincerely appreciate your efforts and constructive comments, which has helped us further improve our work. As the discussion period is approaching the end, we are kindly expecting your valuable feedback to us to ensure your concerns have been addressed satisfactorily. If you still have any additional comments, please let us know, and we are eager to address any remaining issues.
> >
> > Best regards,
> >
> > The authors.

---

> > > ### Comment · Area_Chair_tUu7 · 2025-08-06
> > > **[Urgent] Action Needed on Paper 4013 Rebuttal**
> > >
> > > Dear Reviewer XHw7,
> > >
> > > The authors have provided a detailed rebuttal, which has led other reviewers to raise their scores.
> > >
> > > As the overall evaluation is now positive, your feedback on the rebuttal—and the mandatory acknowledgment—are important and urgent. Please review the rebuttal and provide an update or acknowledgment accordingly.
> > >
> > > Best regards,
> > > Area Chair

---

> > ### Comment · Reviewer_XHw7 · 2025-08-07
> >
> > I appreciate authors' effort on providing detailed benchmarks in a short time, and I think this rebuttal has addressed all my concerns. I will raise my points accordingly. Please make sure to include these additional benchmark in the future version.

---

> > > ### Author Response · Authors · 2025-08-07
> > >
> > > Thank you for your valuable feedback and recognition of our work. We promise to include the latest benchmarks in the updated version.

---

### Note · Authors · 2025-08-14

We sincerely thank all reviewers, ACs, SACs and PCs for their efforts on reviewing and helping improving the manuscript.
### Key contributions
- We provide a convenient framework for integrating **non-paired multi-omics data without relying on any specific pairing information or conventional pairwise transfer** by leveraging a well-designed **beta-VAE architecture with a unified encoder regularized with a joint optimization goal**.
- We conducted **comprehensive evaluations** under different scenarios, demonstrating the excellent performance of scMRDR as well as its **scalability on large-scale datasets and for over 2 modalities**.
- We illustrate the **biological significance** of scMRDR through the downstream tasks of spatial omics integration.
### Rebuttal highlights
- **Comprehensive benchmark**: We added comparison with two recent methods and replaced all metrics with unscaled raw values.
- **Sensitivity analysis**: We added sensitivity analysis on hyperparameter combinations, systematically demonstrating the synergistic effects of different components in our architecture.
- **GLUE's performance**: We explain the performance of GLUE performance on large-scale datasets by conducting comprehensive sensitivity analysis.
- **Downstream significance**: We integrated scRNA, scATAC and spatial transcriptomics, and using the integrated embeddings to impute unmeasured spatial locations. The biological significance of our method is demonstrated through the gain in the detection of spatially highly variable genes.
### Reviewers' feedbacks
- Reviewers acknowledged **the significance of the problem we aim to address, the effectiveness of scMRDR, the comprehensiveness of evaluations, the biological importance demonstrated in downstream tasks, and clarity of writing**.
- **All three reviewers gave positive feedbacks and raised their scores**.
- Reviewers XHw7 and co9u are satisfied with our responses.
- For the remaining concern of Reviewer rgSu: although previous studies attempted to decompose the latent space, they are still limited to partial paired information supervision, stacked encoders, or cross-modal contrastive learning, limiting their ability to the integration of over two modalities with complete non-paired data. In contrast, we **changed the paradigm**, adopting a single unified encoder-decoder beta-VAE architecture for decoupling, with our proposed joint optimization objective as a regularization, making it scalable to more complex scenarios.

---

### Decision · Program_Chairs · 2025-09-17

**Decision:**

Accept (spotlight)

**Comment:**

Summary
This paper introduces scMRDR, a generative framework for integrating unpaired single-cell multi-omics data (e.g., RNA, ATAC, spatial). The model disentangles shared vs. modality-specific representations via a β-VAE, adds adversarial alignment for cross-modality consistency, and introduces an isometric regularization loss to preserve intra-modality structure. A masked reconstruction loss supports incomplete features. The framework is modality-agnostic and demonstrated on tasks including two-omics integration, triple-omics alignment, and large-scale (125k cells) integration, as well as downstream spatial transcriptomics analyses (e.g., imputing spatial locations and detecting spatially variable genes).

Strengths
- Addresses an important and technically challenging problem: unpaired multi-omics integration.
- Provides a well-engineered framework that combines disentanglement, adversarial alignment, and masked reconstruction. The isometric loss is a distinct and effective contribution, helping preserve biological structure (similar ideas exist, e.g. PHATE, but this formulation is novel in this context).
- Demonstrated convincingly on multiple tasks (two-omics, triple-omics, large-scale datasets, and spatial transcriptomics downstream applications). The paper signals clear domain expertise and strong empirical execution.

Weaknesses / Areas for Improvement
- Conceptual novelty is modest, as many components (disentanglement, adversarial alignment, masking) exist in prior work; the main originality lies in the isometric regularization.
- Evaluation focuses on integration benchmarks; broader downstream biological utility (e.g. regulatory discovery, trajectory inference) could strengthen the contribution.

Decision Rationale
Reviewers converged on a positive assessment: the paper is technically sound, clearly presented, and addresses a significant challenge with a carefully designed and scalable framework. While novelty is incremental, the isometric regularization is original and impactful, and the framework convincingly outperforms relevant baselines. The rebuttal clarified concerns around baseline tuning and metrics, and additional downstream analyses further strengthened the case. On balance, the strengths outweigh the weaknesses, and the paper is recommended for acceptance.

Reviewer Discussion
- All reviewers engaged with the rebuttal and raised their scores.
- Consensus: strong execution, modest novelty, but overall positive evaluation.